

# Opinion: New directions in atmospheric research offered by research infrastructures combined with open and data-intensive science

Andreas Petzold[1, 2], Ulrich Bundke[1], Anca Hienola[3], Paolo Laj[3, 4], Cathrine Lund Myhre[5], Alex Vermeulen[6, 7], Angeliki Adamaki[6, 7], Werner Kutsch[7], Valerie Throuret[8], Damien Boulanger[8], Markus Fiebig[5], Markus Stocker[9], Zhiming Zhao[10], Ari Asmi[11]

[1] Forschungszentrum Juelich GmbH, Juelich, 52428, Germany
[2] Institute for Atmospheric and Environmental Research, University of Wuppertal, 42119 Wuppertal, Germany
[3] Finnish Meteorological Institute, Helsinki, 00560, Finland
[4] University Grenoble-Alpes, CNRS, IRD, Grenoble-INP, IGE, 38000 Grenoble, France
[5] The Climate and Environmental Research Institute NILU, Kjeller, 2027, Norway
[6] Lund University, Lund, 22100, Sweden
[7] ICOS ERIC, Helsinki, 00560, Finland
[8] Laboratoire d'Aérologie, CNRS and Université Paul Sabatier Toulouse III, Toulouse, 31400, France
[9] TIB – Leibniz Information Centre for Science and Technology, Hannover, 30167, Germany
[10] University of Amsterdam, Amsterdam, 1012WX, The Netherlands
[11] Research Data Alliance Association (Europe) AISBL, 1040 Etterbeek, Belgium

*Correspondence to*: Andreas Petzold (a.petzold@fz-juelich.de)

**Abstract.**

Acquiring and distributing essential information for understanding global biogeochemical interactions between the atmosphere and ecosystems, and how climate-ecosystem feedback loops may change atmospheric composition in the future is a fundamental pre-requisite for societal resilience in view of climate change. Particularly, the detection of trends and periodicity in the presence of greenhouse gases and short-lived climate-active atmospheric constituents is an important aspect of climate science. Thus, the availability of an easy and fast access to reliable, long-term, and high-quality environmental data is recognized as fundamental for research and for developing environmental prediction and assessment services. In our Opinion Article, we develop the role environmental research infrastructures in Europe (ENVRI RIs) and particularly the atmosphere-centred research infrastructures ACTRIS, IAGOS and ICOS can assume with their capacities for standardised acquisition and reporting of long-term and high-quality observational data, complemented by rich metadata, for the provision of data by open access, and for data interoperability across different research fields including all fields of environmental sciences and beyond. Resulting from these capacities in data collection and provision, we elaborate on the novel research opportunities in atmospheric sciences which evolve from the combination of open-access and interoperable observational data, tools and technologies offered by data-intensive science, and the emerging service ecosystem of the collaboration platform ENVRI-Hub, hosted by the European Open Science Cloud.



## 1    Rationale

Beyond doubt, we have entered the Anthropocene (Steffen et al., 2011) and are now in a period where mankind is the main determinant in the fate of the planet Earth. Natural as well as anthropogenic factors lead to environmental changes on all scales from local to global. Documenting and quantifying these changes are necessary requirements for advancing our scientific understanding of the Earth and its environment, including its complex feedback mechanisms. This documentation and quantification are also needed for developing mitigation and adaptation strategies, for fact-based decision-making, and for the

development of environment-friendly innovations. Furthermore, reliable predictions of environmental change must be based on trustworthy, well-documented observations which capture the entire complexity of the Earth system and the manifold interactions between the atmosphere, the land, and the ocean, and the impacts from and onto life in all its forms. Ultimately, data from all segments of the planet Earth and its environment provide the scientific basis for analysing the physical, biological, and economic processes in the Earth system which are affecting all sectors of society as well as wildlife and biodiversity

(IPCC, 2021, 2023).

The availability of an easy and fast access to reliable, long-term, and high-quality environmental data is recognized as fundamental for research and for developing environmental prediction and assessment services. The resulting requirements for a global climate observing system are now widely discussed and have generated numerous articles on this matter. The

international programme Global Climate Observing System GCOS (2010, 2016) coordinated by the World Meteorological Organisation (WMO) introduced the "essential climate variables" (ECVs) to provide a science-based framework for coordinated climate observations; see more on that topic in Section 3.2. Founded on this framework, Weatherhead et al. (2018) proposed one potential approach towards the observing system of the future. In this article, Weatherhead and co-workers note that for each Grand Challenge identified by the World Climate Research Program (WCRP), observations are needed for long-

term monitoring, process studies and forecasting capabilities. The proposed observing system should include satellites, ground-based and in situ observations as well as potentially new, unidentified observational approaches.

Carmichael et al. (2023) from the WMO Global Atmosphere Watch (WMO GAW) programme, discuss critically the requirements for a long-term sustained and research-driven global observation infrastructure for atmospheric composition.

They identify an "urgent need to move from opportunities-driven one-component observations to more systematic, planned multifunctional infrastructure, where the observational data flow is coupled with Earth system models to serve both operational and research purposes". Kulmala and co-workers discuss in a series of articles the needs for an integrated global climate observing system with specific approaches for different study targets and environments (Kulmala, 2018; Kulmala et al., 2023a; Kulmala et al., 2023b). Despite their different foci, all articles conclude that comprehensive and open data are essential to meet

Environmental Grand Challenges (Kulmala et al., 2021), and that strong efforts have to be put on the development of standards and tools to advance the use of observations and provide open access and interoperable reliable data (Carmichael et al., 2023).





To cope with the scale and complexity of the threats posed to our planet by climate change, environmental sciences need to integrate excellent curiosity-driven science in the traditional fields of Earth system sciences with science focusing on human interactions and societal grand challenges. The task is huge, but the means are there. The Strategy Working Group on
Environment of the European Strategy Forum on Research Infrastructures (ESFRI) has identified in their recent Landscape Analysis of the Environment Domain (ESFRI, 2021b), that European Environmental Research Infrastructures (ENVRI RIs) in atmospheric sciences, jointly with the other branches of environmental sciences, are well positioned to tackle the challenges of climate change at all levels.

In our Opinion Article, we develop the role ENVRI RIs in Europe can assume with their capacities for standardised acquisition and reporting of observation data complemented by rich metadata, for the provision of reliable, long-term, and high-quality environmental data by open access, and for data interoperability across environmental research fields. Resulting from these capacities in data collection and provision of data and services, we will elaborate on the novel research opportunities and methods which evolve from the combination of open-access and interoperable environmental data and tools and technologies
offered by data-intensive science. It should be noted that the focus of our Opinion Article is on atmospheric sciences, although most of the conclusions drawn apply to other areas of environmental research as well.

## 2    Setting the Stage

Recognizing that, given the scale, scope, and high level of complexity of environmental research, the in-depth investigation of the effects of climate change and their impacts on human health, food security and other aspects of the United Nations
Sustainable Development Goals (SDG) requires a critical mass in terms of scientific capacities and resources, the European Commission has initiated the development and implementation of environmental research infrastructures at the European level. Guided by the European Strategy Forum on Research Infrastructures (ESFRI), a roadmap process was started in the early 2000 years to establish pan-European environmental research infrastructures, focusing on the knowledge needed for the promotion of sustainable management of the natural and human environment and its resources (ESFRI, 2021a); visit the website of the
community of Environmental Research Infrastructures (ENVRI) for detailed information.

This new type of scientific infrastructures is best suited for the large-scale and long-term production of high-quality environmental observational data and research products, as well as for operating tools and technologies necessary to handle the data volumes and enable data-intensive research throughout the environmental sciences. Astronomy and high-energy
physics can serve as role models for this kind of infrastructures (Bell et al., 2009), but with the essential difference that environmental observations always require the operation of largely distributed measurement networks together with the handling of complex data streams from highly diverse sources.



More generally speaking, the discovery of scientific knowledge has developed from experimental sciences, conducted by field
observations and experiments, through theoretical sciences shaped by models and mathematical generalizations, to the
simulation of phenomena which are too complex for analytical solutions (Gray, 2009). Now we have entered the age of the
fourth paradigm of data-intensive science, which is described as collaborative, networked and data-driven (Bell et al., 2009;
Lynch, 2009). European environmental research infrastructures are perfectly prepared for this challenge since they combine
research-driven observations with comprehensive capabilities for sustainable data management and stewardship.


The integration of data-intensive sciences and environmental sciences is critical for improving our current understanding of
the major challenges to our planet, such as climate change and its impacts on the Earth system, our ability to respond to and
predict natural hazards, and our understanding and prevention of ecosystem loss. This effort requires the unimpeded use of
multidisciplinary data from the different spheres of the Earth system, with Findability, Accessibility, Interoperability, and Re-
usability of the digital (meta)data (FAIR Principles: Wilkinson et al., 2016) as an absolute prerequisite, embedded in a culture
of open scientific cooperation as stated by the European Commission in their European Open Science Cloud (EOSC)
Declaration (EC, 2017) and outlined by the Organisation for Economic Cooperation and Development (OECD; Dai et al.,
2018).

The scientific power of this kind of production and use of environmental data is impressively demonstrated by the series of
reports published by the Intergovernmental Panel on Climate Change (IPCC); see, the Assessment Report 2021 (IPCC, 2021)
as the latest product. The urgency of further integrating scientific, political and societal efforts in view of the rapidly advancing
climate change is presented in the current IPCC Synthesis Report (IPCC, 2023).

Particularly, the scientific importance of long-term greenhouse gas observations as conducted since many years by the ENVRI
RI ICOS, and since recently, by IAGOS, has been confirmed in the Scientific Background document on the 2021 Physics
Nobel Prize which went to Klaus Hasselmann, Syukuro Wanabe and Giorgio Parisi "For ground breaking contributions to our
understanding of complex physical systems". In Chapter IV, the Nobel Committee writes: "From the perspective of laboratory
science, using experimental measurements to test theory is such a self-evident step in the scientific method that it goes without
saying. However physical cosmology and physical climatology are observational sciences – practitioners observe that which
nature allows."

Another success story of the cross-discipline use of environmental observational data in Earth science is the European
Copernicus programme as the Earth observation component of the European Union's Space programme. In this context,
environmental observational data from ground-based, airborne, or seaborne platforms serve as the in-situ (non-space)
component of the integrated global Earth observation programme outlined by GCOS and supply essential data for the
calibration and continuous validation of spaceborne instruments. For 25 years Copernicus has now provided information



services free and openly accessible to users that draw from satellite-borne Earth observations and in-situ observations, with the pan-European environmental research infrastructures being one of their prime data and service contributors.


Besides research on climate change, the scientific field of atmospheric chemistry with its links to climate change, and the resulting interaction of air quality and human health constitute another highly interdisciplinary research area in the current agenda of atmospheric sciences (von Schneidemesser et al., 2015; The Lancet Planetary, 2020; Romanello et al., 2022). A series of recent past-COVID-19 studies on the effects of reduced anthropogenic emissions on atmospheric chemical

composition (Gkatzelis et al., 2021; Sokhi et al., 2021) impressively illustrate the growing need but also the large opportunities of scientific approaches crossing the traditional boundaries of scientific disciplines and methods. The studies utilise long-term and highly-quality observations of atmospheric composition on a global scale, combined with statistical approaches for identifying patterns in atmospheric composition change which can be traced back to modified emission behaviours in different parts of the world.


Furthermore, the study by Gkatzelis et al. (2021) demonstrates the added value gained from the use of state-of-the-art tools of data-driven science for creating knowledge in atmospheric sciences. This study combines a traditional scientific review approach with the use of an openly accessible relational data base management system which holds the data from the publications included in the review (JUELICH-COVID-19). The approach allows the construction of a living review because

new publications and data can easily be incorporated in the study and updated figures are produced directly from the database. The last database update originates from 26 January 2023 (version 5), adding 64 new publications with 930 new measurements.

Operational European Research Infrastructures with their capacities for long-term continuous provision of observational data and services about air quality and atmospheric composition are comprehensively prepared to assume their role in this important

field of cross-disciplinary research, jointly with life sciences and social sciences. The currently ongoing European research projects entitled Research Infrastructures Services Reinforcing Air Quality Monitoring Capacities in European Urban & Industrial AreaS (RI-URBANS) and ICOS Cities (ICOS-Cities) are driving the development and implementation of air quality and greenhouse gas monitoring systems that will allow European cities as well as health administrations and agencies to effectively mitigate the impact of poor air quality on human health and to support climate action.





### 3 Sustainable Infrastructures for Monitoring Atmospheric Composition

#### 3.1 Environmental Research Infrastructures in Atmospheric Sciences

##### 3.1.1 The Community of Environmental Research Infrastructures

By definition in Article 2 (6) of the Regulation (EU) No 1291/2013 of 11 December 2013 (EC, 2013), " 'research infrastructures' mean facilities, resources and services that are used by the research communities to conduct research and foster
innovation in their fields. Where relevant, they may be used beyond research, for example for education or public services. They include major scientific equipment or sets of instruments; knowledge-based resources such as collections, archives, or scientific data; e-infrastructures such as data and computing systems and communication networks; and any other infrastructure of a unique nature essential to achieving excellence in research and innovation. "

Research infrastructures offer one solution to make science more effective and sustainable in the complex European funding landscape. They enable an organized, fair, and transparent system to share knowledge and resources, and in doing so, they contribute to the pooling of data, facilities, and equipment, thereby avoiding unnecessary duplication of effort. By making high-quality facilities, resources, and services available to everyone, research infrastructures ensure that science is driven by excellence and not by the research capacity of individual countries, economic sectors, or institutions. They also ensure that
this excellence is aimed at solving bottlenecks, pushing forward the frontiers of scientific disciplines, and enabling transformative technological development.

The science cluster of ESFRI-listed environmental research infrastructures (ENVRI) in Europe was formed as a strong community of principal producers and providers of environmental research data and research services in Europe. For most of
the ESFRI-listed RIs, access to their services is not simply limited to digital (data) services. Moreover, physical, and remote access to the different types of research facilities (i.e., central laboratories, reference instruments, research platforms) are another integral part of their service portfolio and fulfilment of their research and innovation strategies. The science cluster of environmental RIs is one out of five science clusters of the ESFRI list of RIs. The other clusters focus on Energy, Health & Food, Physical Sciences & Engineering, and Social and Cultural Innovation; see the latest ESFRI Roadmap Report for more
information (ESFRI, 2021a).

The ENVRI cluster covers all segments of the Earth and its environment, from the atmosphere, water, and solid earth spheres to the biosphere and finally to the anthroposphere. According to the ESFRI Roadmap 2021 (ESFRI, 2021a), the cluster consists of the ESFRI Landmarks ACTRIS, EISCAT_3D, EMSO ERIC, EPOS ERIC, EURO-ARGO ERIC, IAGOS AISBL, ICOS
ERIC, and LifeWatch ERIC, and the ESFRI projects DANUBIUS-RI, DiSSCo, and eLTER RI, with AnaEE ERIC from the Health & Food domain contributing to the sciences on the biosphere. SIOS contributes as a local multi-domain research observatory, situated in Svalbard, Spitzbergen, and is focusing on Arctic research.



In the past decade, through the European Commission cluster projects ENVRI (2011-2014), ENVRIplus (2015-2019), and ENVRI-FAIR (2019-2023), the ENVRI RIs have created a robust conceptual and technical framework based on the FAIR
principles, that empowers the ENVRI cluster to provide discipline or domain-oriented (e.g., atmosphere, marine, land system) or thematically/methodologically oriented (e.g., carbon cycle) services in support of climate change research in all its facets (Petzold et al., 2019). This framework includes common standards and policies for the data life cycle, including cataloguing, curation, provenance, and service provision within the ENVRI cluster, with specific consideration of the FAIR principles including interoperability solutions and open standards, e.g., on metadata description. Open access to data sets generated,
hosted, and curated by the ENVRI RIs is another integrating element of this framework, which is currently realised at RI level, but will be lifted to cluster level once cross-RI authentication and authorisation infrastructures (AAI) are in place. The numerous aspects tackled during these joint efforts are described in detail in the contributions to the book entitled "Towards Interoperable Research Infrastructures for Environmental and Earth Sciences", edited by Zhao and Hellström (2020).

### 3.1.2     Atmosphere-centred Research Infrastructures

The ENVRI landscape of atmosphere-centred environmental research infrastructures (ATMO-RIs) is illustrated in Figure 1, whereas Table 1 compiles brief descriptions of the ATMO-RIs, with the Aerosol, Clouds, and Trace Gases Research InfraStructure (ACTRIS), the In-service Aircraft for a Global Observing System (IAGOS) and the Integrated Carbon Observation System (ICOS) - Thematic Centre Atmosphere - performing research on the chemical composition of the atmosphere in a changing climate. The Svalbard Integrated Arctic Earth Observing System (SIOS) is organising multi-domain
observation platforms at Svalbard archipelago in the Arctic region, with atmosphere-related research data largely provided via ACTRIS and ICOS, while EISCAT_3D is concentrating on the dynamics of the mesosphere. Thus, the ATMO-RI landscape consists of three RIs targeting solely atmospheric research topics (ACTRIS, EISCAT, IAGOS), and two multi-domain RIs with an atmospheric component (ICOS, SIOS). In the remaining part of our Opinion Article, we will concentrate on the role of ATMO-RIs providing continuous data on atmospheric composition from a European-wide network of surface-based
observation stations (ACTRIS, ICOS) and a fleet of passenger aircraft equipped with scientific instrumentation (IAGOS).



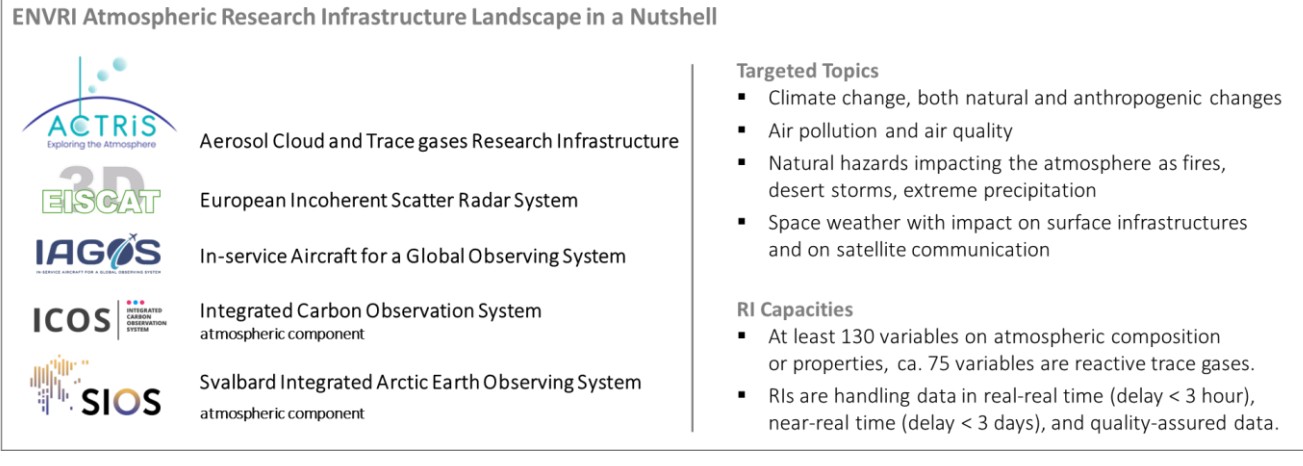

**Figure 1.** The landscape of environmental research infrastructures in Europe (ENVRI) in atmospheric sciences in a nutshell.

**Table 1.** ESFRI Environmental Research Infrastructures of the Atmospheric Sciences.

| | |
|---|---|
| ACTRiS<br>Exploring the Atmosphere<br>Aerosol, Clouds, and Trace Gases Research InfraStructure<br><br>https://www.actris.eu/ | ACTRIS is an ESFRI landmark. The associated ACTRIS-ERIC was created in 2023, with the commitment of 17 countries to join, representing more than 100 research performing organisations. ACTRIS coordinates activities for documenting concentrations, understanding processes, and quantifying impacts of short-lived atmospheric constituents like reactive trace gases and aerosol particles, and here especially black carbon, on Earth's climate, air quality, human health, and ecosystems. It operates a network of more than 100 facilities (ground-based observation sites and experimental platforms), supported by 6 Central Laboratories (Topical Centres) and one Data Centre (ACTRIS-DC). |
| IAGOS<br>IN-SERVICE AIRCRAFT FOR A GLOBAL OBSERVING SYSTEM<br><br>https://www.iagos.org/ | IAGOS is an ESFRI landmark established in 2014 as an International not for profit Association under Belgian law (IAGOS-AISBL) with partners from France, Germany, and the United Kingdom. IAGOS combines the expertise of scientific institutions with the infrastructure of civil aviation to provide essential data on climate change and air quality at a global scale. Data cover ECVs air temperature, $O_3$ $H_2O$, CO, $NO_x$, $CO_2$, $CH_4$, aerosols and clouds, and are delivered both as tropospheric profiles from climb-out and descending phases and in the upper troposphere and lowermost stratosphere (UTLS) at cruise. The data are measured aboard 10 passenger aircraft on a day-to-day basis and by one flying laboratory with lesser coverage on one passenger aircraft. The IAGOS Data Centre provides open access to data and data products. |
| | ICOS is an ESFRI landmark established as an ERIC (ICOS-ERIC) in 2016, with 14 countries participating and a community of more than 80 Research performing organisations. ICOS-based |



| | |
|---|---|
| 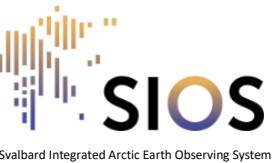<br>https://www.icos-cp.eu/ | knowledge supports policy- and decision-making to combat climate change and its impacts. ICOS provides standardised and open data from more than 140 measurement stations across European countries. The stations observe greenhouse gas concentrations in the atmosphere as well as carbon fluxes between the atmosphere, the land surface, and the oceans. ICOS data quality and measurement procedures are controlled by its three Thematic Centres and data and data-products made available through the Carbon Portal. |
| 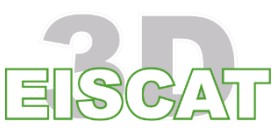<br>https://sios-svalbard.org/ | The Svalbard Integrated Arctic Earth Observing System is a distributed research infrastructure organised as a consortium of 29 member institutions from 10 countries. Their common goal is to establish a cooperating and transparent research infrastructure which will give better estimates of the future environmental and climate changes in the Arctic. SIOS focuses on processes and their interactions between the different spheres, i.e., biosphere, geosphere, atmosphere, cryosphere, and hydrosphere. The core observational programme of SIOS provides the research community with systematic observations that are sustained over time, yet dynamic enough to be adapted as new methods appear or society poses new questions. |
| https://eiscat.se/ | EISCAT 3D is an international research infrastructure, using radar observations and the incoherent scatter technique for studies of the atmosphere and near-Earth space environment above the Fenno-Scandinavian Arctic as well as for support of the solar system and radio astronomy sciences. The radar system is designed to investigate how the Earth's atmosphere is coupled to space, but it will also be suitable for a wide range of other scientific targets for e.g., space weather forecasts, detecting space debris and climate research. |


*Harmonising and Securing the Landscape of Atmospheric Observations*

Historically, the ATMO-RIs have developed from independent research projects which started their integration processes with the emerging ESFRI Roadmap Process in the early 2000 years. All predecessor projects focused on different research targets

but in similar research fields. ACTRIS brought together ground-based cloud remote sensing measurements (CloudNET), the European supersites for atmospheric aerosol research (EUSAAR), a European aerosol research lidar network (EARLINET), and the European atmospheric simulation chambers (EUROCHAMP). IAGOS merged two projects on the measurements of ozone, water vapour and other trace species by a fleet of in-service Airbus aircraft (MOZAIC; Marenco et al. (1998)), or by means of one instrumented airfreight container (CARIBIC; Brenninkmeijer et al. (1999)) into one infrastructure on

atmospheric observation by civil passenger aircraft (Petzold et al., 2015; Thouret et al., 2022). Finally, ICOS was built around the scientific target of continuous, long-term observations of the carbon cycle, including the quantification of greenhouse gas (GHG) emissions, sinks, and their impacts on Earth systems (Heiskanen et al., 2022). Predecessor projects are numerous



(> 25), and Fig.1 in Franz et al. (2018) illustrate the fragmentation of observation networks in Europe which then developed into ICOS. While ACTRIS, IAGOS and ICOS infrastructures are characterised by strongly distributed observation networks,
EISCAT and SIOS have emerged from single-sited activities.

Looking at the evolution of the ATMO RIs, one important achievement of the research infrastructure process becomes immediately apparent, namely the integration of a formerly scattered landscape of multiple observation networks into few research infrastructures. The direct benefits of this consolidation process were joining resources of all kinds, reducing
duplication of work, and offering data of previously unknown quality and complexity. Before the consolidation phase, data have been locked in silos and were poorly standardised over decades, hampering scientific progress severely. During the consolidation process, the ATMO-RIs have developed cutting-edge standardisation for their observations and implemented highly efficient data life cycle management systems, leading to timely, open, and FAIR data access by scientists all over the world. Tackling the next step, the ATMO-RIs joint forces in the European Union's Horizon 2020 research and innovation
programme "Sustainable Access to Atmospheric Research Facilities" (ATMO-ACCESS) for establishing a comprehensive and sustainable framework for access to distributed atmospheric RIs, ensuring integrated access to and optimised use of the services they provide.

Another major achievement of the ATMO-RIs which deserves explicit mentioning is the conservation of long-term
atmospheric observation stations and time series which were in danger during the transition process from research projects to research infrastructures. The detection of trends and periodicity in the presence of greenhouse gases and short-lived climate-active atmospheric constituents is an important aspect of climate science. An accurate description of trends relies heavily on the ability to place the measurements of all kinds of climate-active atmospheric constituents in a historical context, i.e., to compare measurements against measurements from the same location in preceding years and decades. From a scientific
perspective, longer timeseries reduce the uncertainties in the interpretation of current measurements. The length of timeseries which are produced by the different ATMO-RIs is therefore a good indicator of how well these data meet the requirements of climate scientists.

To give examples, we first may refer to the long-term in-situ observations of ozone and water vapour in the upper troposphere
and lowermost stratosphere by passenger aircraft equipped with MOZAIC instrumentation (Marenco et al., 1998), starting in 1994. These long-term data could not only be secured but also subjected to the most modern data quality assurance, and successfully continued after the transition of MOZAIC to IAGOS was completed. Today these data sets cover the climatologically relevant length of 30 years and are widely used, e.g., in the IPCC reports, in the Tropospheric Ozone Assessment Report (Schultz et al., 2017; Tarasick et al., 2019) and in climatological studies (Petetin et al., 2016; Gaudel et al.,
2018; Gaudel et al., 2020; Petzold et al., 2020). The representativeness of the IAGOS airborne measurements in the lower troposphere was explicitly investigated to ensure the applicability of these airborne data sets in the context of air quality studies



(Petetin et al., 2018). Without the implementation of IAGOS as a research infrastructure, neither the historical data would be available, nor the continuation of observations would have been possible.

The second example focuses on the measurements of greenhouse gases and the carbon cycle that are a core element in the scientific fields of ecology and evolution, environmental sciences, atmospheric sciences, forestry, and agronomy. ICOS provides long-term, high-quality observations that follow (and cooperatively set) the global standards for the best possible quality data on the atmospheric composition for greenhouse gases (GHG), greenhouse gas exchange fluxes measured by eddy covariance and $CO_2$ partial pressure at water surfaces. Next to these main data products ICOS also produces observations of 275 many ancillary variables, using the same highest quality standards. The ICOS measurement infrastructure is for a large part build from existing measurement stations which are updated to provide measurements that meet ICOS standards. Data which were collected before ICOS measurement protocols were put in place, have been secured physically and with respect to data quality, and contribute now to the ICOS record of long-term data sets.

Of ICOS' 134 measurement stations 101 stations provided data on the length of timeseries held by them. Data from these 101 stations describe how long they have been operational, or, in some cases, how long the station has been collecting measurements that are relevant to ICOS. The average length of timeseries across all domains is 11 years, and this is evenly spread between the ecosystem domain (average 11 years), atmosphere domain (12 years) and ocean domain (11 years); see Figure 2. Timeseries from atmosphere stations tend to be the longest, which likely reflects the overlap between historical 285 atmosphere measurements and current variables being measured by ICOS. Details are given in the latest ICOS Assessment Report (Belle et al., 2018).

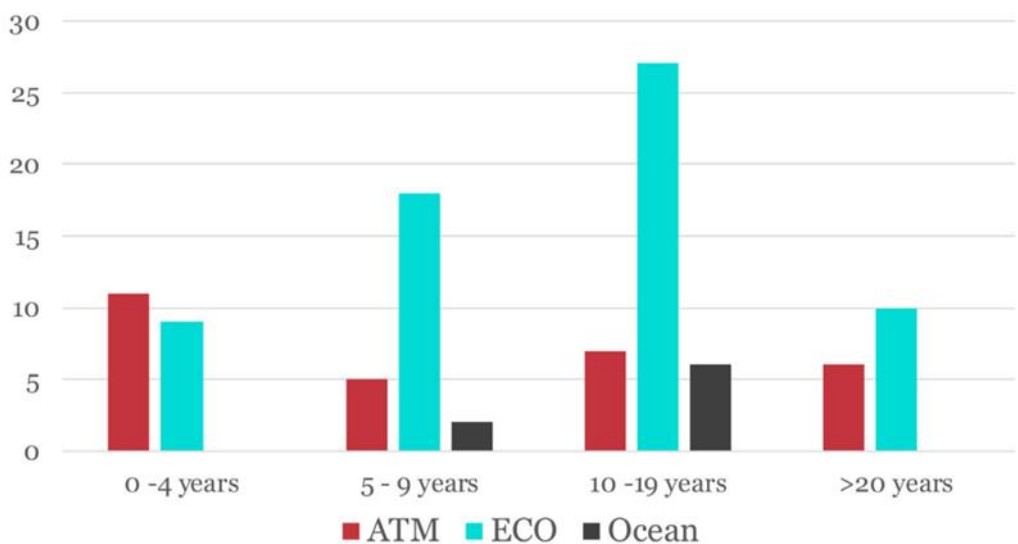

**Figure 2.** Number of measurement stations (y-axis) and length of timeseries (x-axis) held by ICOS measurement stations in 2018, for the Atmosphere (ATM), Ecosystem (ECO), and Ocean domains (Source: Belle et al., 2018).





The third example is the aerosol ECV "Aerosol Single Scattering Albedo" retrieved from ACTRIS measurements. ACTRIS is covering observation of short-lived constituents of the atmosphere, using in situ and remote sensing methods located on the Earth surface. The series of ACTRIS projects, leading to the establishment of the ACTRIS ERIC, started in 2011, but predecessor projects date back to the year 2000, single time series of observations started already in 1995. Figure 3 shows an example of a long timeseries of aerosol particle scattering and absorption coefficient observed at the Ispra station in Northern

Italy (upper and mid panel respectively). From data with both high quality and high time resolution, the aerosol particle single scattering albedo can be calculated, an ECV necessary for assessing the direct aerosol climate effect. The single scattering albedo is shown in the lower panel over the period 2009 – 2023.

Overall, long-term observations of climate-relevant aerosol properties by ACTRIS became an indispensable ingredient of the

global climate observation system and have trigged multiple scientific studies on the long-term aerosol impact on climate (e.g., Zanatta et al., 2016; Collaud Coen et al., 2020; Laj et al., 2020). Trends are detectable in the data, possible only due to collection of data with consistent quality and operating procedures over long time periods.

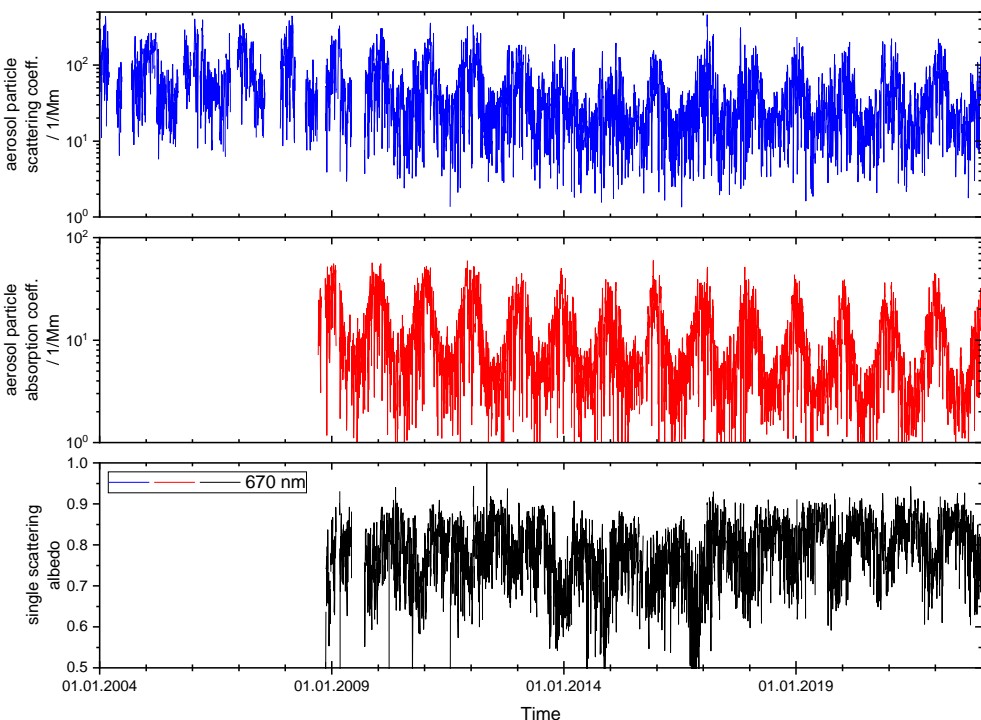

**Figure 3.** Time series of aerosol particle scattering (top panel; Putaud and Martins dos Santos (2022b)) and absorption coefficients (middle panel; Putaud and Martins dos Santos (2022a)), observed at the Ispra station in Northern Italy associated to ACTRIS and ACTRIS pre-projects, all at 670 nm wavelength. The data displayed in the two upper panels form the basis for calculating the aerosol particle single scattering albedo, which is declared as essential climate variable (ECV).





*Technical Achievements*

Technically, the ATMO-RIs have definitely pushed the borders of what can be observed. The measurements are highly standardised, connected to unified calibration systems, partially available in near-real time and well distributed over the European continent, or even globally. Measurement methods deployed by the ATMO-RIs follow published common specifications and protocols and use synergies among the RIs as efficient as possible. The data is quality controlled and processed at dedicated data centres, using open and published processing chains. The RI specific calibration laboratories

provide the stations with working standards and analyse flask samples for trace gas concentrations. All data from raw data up to the final quality controlled (averaged) data is openly accessible from the RI data portals (ACTRIS and IAGOS Data Portals, ICOS Carbon Portal).

Today, ICOS provides the densest and most integrated observational network on greenhouse gases worldwide, IAGOS is a

globally unique observation infrastructure for atmospheric trace gases, aerosol and cloud particles which reaches parts of the atmosphere not accessible for in-situ observations by other platforms, and ACTRIS constitutes a research infrastructure on short-lived atmospheric constituents like reactive trace gases, aerosols and clouds, which investigates the processes leading to the variability of these constituents in the natural environment and in controlled atmospheres of unprecedented complexity. The strategic role of the European complementary RIs in the atmospheric sciences (the ATMO-RIs) has been highlighted in

the recent IPCC report (IPCC, 2021; see Section 1.5.1.1 Major Expansions of Observational Capacity), noting the key contribution of ICOS, IAGOS, and ACTRIS. These three ATMO-RIs are also essential data providers to the Copernicus services in the areas of Atmosphere Monitoring (CAMS; Wagner et al. (2021); Peuch et al. (2022)) and Climate Change (C3S; Thepaut et al. (2018)) as well as to the WMO Global Atmosphere Watch Programme (GAW; Carmichael et al. (2023)) and play leading roles in the evolving WMO Greenhouse Gas Monitoring Infrastructure.

**3.2   The Science-based Framework of Essential Climate Variables**

The Earth atmosphere is a complex media with thousands of chemical species interacting with their environment. Consequently, monitoring the atmospheric composition comprehensively in the long term is impossible. To transform the challenging task of integrated Earth observation into a concept towards a global climate observation system, WMO/GCOS has defined a set of global climate indicators (WMO-GCI), which stretches beyond the boundaries of traditional scientific

disciplines like, e.g., atmospheric sciences, ocean sciences, or biology. WMO requests the continuous observation of these indicators for monitoring the state of Earth's climate. However, this undertaking requires a largely interdisciplinary approach. In Figure 4, we propose a modified set of Global Climate Indicators, with short-lived climate forcers having been added to the original set, given the results of the latest IPCC Report (IPCC, 2021).



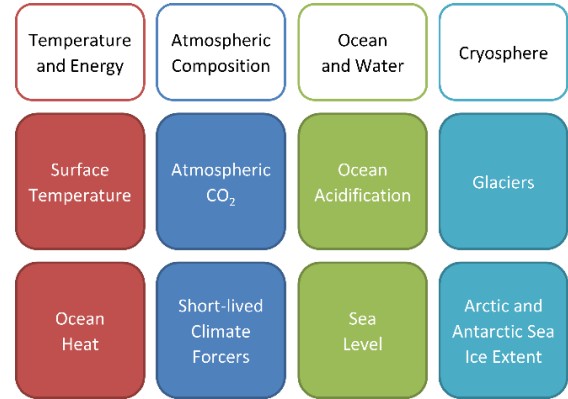


**Figure 4.** Global Climate Indicators; adapted from WMO/GCOS (WMO-GCI); short-lived climate forcers have been added to the original set, given the results of the latest IPCC Report (IPCC, 2021).


To further develop this approach towards a global climate observation system, WMO has introduced the concept of essential atmospheric variables, defined as a physical, or chemical variable or a group of linked variables that critically contributes to the characterization of the atmospheric composition. Obviously, the essential variables, or group of variables, and the user-driven requirements may vary, depending on the type and the area of applications. The observables to be recorded for the

different application areas are defined more detailed by WMO through the process of Rolling Review of Requirement in the Observing Systems Capability Analysis and Review Tool (WMO-OSCAR).

Among the twelve application areas of essential variables, two are critically relevant to the work of atmospheric research infrastructures:

1. Atmospheric Climate Forecasting and Monitoring defines the requirements for climate observations, to detect, model and assess climate change, support adaptation to climate change and monitor the effectiveness of policies for mitigating climate change. Essential Climate Variables (ECV) for the atmosphere, the oceans and terrestrial systems (Bojinski et al., 2014) are identified for this application area. As is defined by Bojinski et al. (2014), an ECV is a physical, chemical, or biological variable or a group of linked variables that critically contributes to the

360        characterisation of Earth's climate. ECV datasets provide the empirical evidence needed to understand and predict the evolution of climate, to guide mitigation and adaptation measures, to assess risks and enable attribution of climatic events to the underlying causes, and to underpin climate services. The current list of ECVs is specified by the Global Climate Observing System (GCOS, 2010). The updated requirements for the observation of ECVs were recently published by GCOS (2022b).






2.  Atmospheric Composition Forecasting and Monitoring defines requirements related to evaluating and analysing changes (temporally and spatially) in atmospheric composition regionally and globally to support treaty monitoring, climatologies and re-analyses, assessing trends in composition and emissions/fluxes, and to better understand processes, using data of controlled quality. The essential variables (EV) for this application area are defined under
the Global Atmosphere Watch (GAW) program of WMO. Essential variables are needed to understand and predict the evolution of atmospheric composition as a whole.

Both EV and ECVs are identified based on their relevance for the specific application area, but also on the fact that observation of the variable on a global scale is technically feasible using proven, scientifically understood methods. While the list of
essential variables for the two applications areas overlaps to some extent, requirements in terms of timeliness, spatial and temporal definition and/or quality objectives may differ.

The recommended essential measurement variables in GAW and GCOS are listed in Table 2a and b, respectively. A complete list of requirements can be found through the OSCAR table. Table 2b lists also the ATMO-RIs covering the respective
parameters. ECVs based on total column densities of specific molecules are covered by a variety of satellite-borne instruments and by ground-based remote sensing observational networks such as the Total Carbon Column Observing Network (TCCON; Wunch et al. (2011)),  the Network for the Detection of Atmospheric Composition Change (NDACC; De Maziere et al. (2018)), the federated network for aerosol characterisation AERONET (Holben et al., 1998), or the global ozonesondes network (Tarasick et al., 2021).


**Table 2a**. Essential variables for the application area Atmospheric Composition Forecasting and Monitoring as defined in the GAW Implementation Plan 2024-2029.

| | |
|---|---|
| Greenhouse gases | Carbon dioxide $CO_2$ |
| | Methane $CH_4$ |
| | Nitrous oxide $N_2O$ |
| | Halogenated compounds and $SF_6$ |
| Aerosols | Mass concentration (fine, coarse) |
| | Major chemical components |
| | Aerosol optical depth |
| | Optical properties including light absorption and scattering |
| | Aerosol number and size |
| | Vertical distribution of aerosol backscattering and extinction |




| | | |
|---|---|---|
| | Aerosol hygroscopicity measurement | |
| Reactive gases | Surface and tropospheric ozone | |
| | Carbon monoxide | |
| | Volatile Organic Compounds | |
| | Nitrogen oxides | |
| | Sulphur dioxide | |
| | Molecular hydrogen | |
| Atmospheric total deposition: | pH of wet deposition | |
| | Conductivity of wet deposition | |
| | Alkalinity of wet deposition | |
| | Chemical composition of wet deposition | |
| | Dry deposition | |
| Ozone & UV radiation | column (total) ozone | |
| | Ozone vertical profiles with a focus on the stratosphere and upper troposphere | |
| | UV-A/UV-B | |
| | Ozone depleting substances | |

**Table 2b.** Essential climate variables for the application area Atmospheric Climate Forecasting and Monitoring as defined in the GCOS Implementation Plan 2022 (GCOS, 2022a), and the ATMO-RIs providing the observations.

| Atmosphere ECV | | ATMO Research Infrastructures |
|---|---|---|
| Cloud Properties | Cloud cover | ACTRIS |
| | Cloud liquid water path | ACTRIS |
| | Cloud ice water path | ACTRIS |
| | Cloud drop effective radius | ACTRIS |
| | Cloud optical depth | ACTRIS |
| | Cloud top temperature | not covered by ATMO-RIs |
| | Cloud top height | not covered by ATMO-RIs |
| Carbon dioxide, methane and other greenhouse gases | $N_2O$ mole fraction | ICOS, IAGOS (in-situ Trop./LMS) |
| | $CO_2$ total column | not covered by ATMO-RIs |
| | $CO_2$ mole fraction | ICOS (surface), IAGOS (in-situ Trop./LMS) |
| | $CH_4$ total column | not covered by ATMO-RIs |
| | $CH_4$ mole fraction | ICOS (surface), IAGOS (in-situ Trop./LMS) |
| Ozone | $O_3$ VMR in the troposphere | IAGOS (in-situ) |





| | O₃ VMR in the UTLS | IAGOS (in-situ) |
|---|---|---|
| | O₃ VMR in the middle/upper stratosphere | not covered by ATMO-RIs |
| | O₃ stratospheric column | not covered by ATMO-RIs |
| | O₃ tropospheric column | IAGOS (potentially from tropospheric profiles) |
| | O₃ total column | not covered by ATMO-RIs |
| Precursors (supporting the aerosol and ozone ECVs) | CO total column | not covered by ATMO-RIs |
| | CO tropospheric column | IAGOS (potentially from tropospheric profiles) |
| | HCHO total column | ACTRIS (surface) |
| | SO₂ total column | not covered by ATMO-RIs |
| | NO₂ total column | IAGOS (potentially from tropospheric profiles) |
| | CO mole fraction | ACTRIS (surface), IAGOS (in-situ Trop./LMS) |
| | NO₂ mole fraction | ACTRIS (surface), IAGOS (in-situ Trop./LMS) |
| Aerosol properties | Aerosol $\sigma_{ext}$ vertical profile (Trop.) | ACTRIS (Lidar), IAGOS (in-situ) |
| | Aerosol $\sigma_{ext}$ vertical profile (Strat.) | ACTRIS (Lidar) |
| | Multi-wavelength AOD | ACTRIS (Lidar), IAGOS (in-situ Trop.) |
| | Chemical composition of aerosol | ACTRIS (surface), IAGOS (in-situ Trop./LMS) |
| | Number of cloud condensation nuclei | ACTRIS (surface) |
| | Aerosol Single Scattering Albedo | ACTRIS (surface) |
| | Aerosol Number Size Distribution | ACTRIS (surface), IAGOS (in-situ Trop./LMS) |

Abbreviations and symbols: Trop. = Troposphere; UT = upper troposphere, LMS = lowermost stratosphere, VMR = volume mixing ratio; $\sigma_{ext}$ = aerosol light extinction coefficient.


Concluding from the latest landscape analysis of the Environment Domain (ESFRI, 2021b), ENVRI RIs and ATMO-RIs in particular, have been indispensable for monitoring and understanding changes in essential variables EVs and ECVs over several decades. Given the great variety and complexity of observational data provided by all ENVRI RIs, the science-based framework of EVs and ECVs offers a concept and guideline for the implementation of tools and standards on the data and

metadata level, that concentrates on key variables covered by multiple ENVRI RIs and allow searchability, accessibility and interoperability of these key ECV data, with the final goal of enabling cross-RI exploitation of data.

As one example for this framework approach, we focus on the atmosphere-related ECVs which are acting on the Earth radiation budget, as defined in the recent IPCC AR6 report (IPCC, 2021) As is illustrated in Figure 5, the ATMO-RIs cover all ECVs

related to atmospheric composition and temperature and water vapour for the upper-air atmosphere, with the latter provided



by IAGOS and ACTRIS for the tropospheric column. As mentioned previously, ATMO-RIs have a broader focus than the ECVs, but this set of parameters forms the common basis between the involved RIs. Enabling searchability, findability and interoperability of these ECV data at all ATMO-RIs allows the scientific exploration of ECV data from all contributing data providers and finally the provision of complex scientific analyses on multiple-ECV data sets. Such complex analyses are out

of reach today or can be accomplished only with large efforts during the collection of data from the individual data providers.

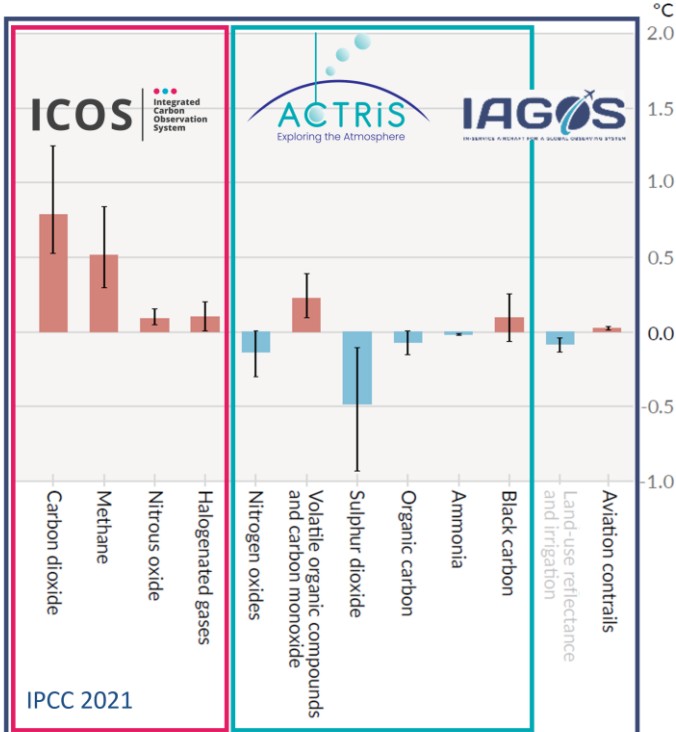

**Figure 5.** Coverage of essential climate variables of the upper atmosphere and atmospheric composition by ATMO-RIs to;
adapted from IPCC (2021). Only the parameter "Land-use reflectance and irrigation" is not covered.

To illustrate with more detail the necessary steps towards interoperable and harmonized ECV data, we concentrate on the most relevant ECVs for climate change which are temperature and greenhouse gas atmospheric abundance. Hundreds of in-situ
stations and satellite systems deliver observational data on these ECVs that need to be calibrated, processed (data reduction), interpolated, re-gridded, and quality controlled in complex workflows to higher-level high-quality data sets that then again are used in model systems to derive other ECV data that answers the relevant policy questions. Traceability, transparency, and information on the data quality through the whole value chain are essential for using this data in relevant societal evaluations





and decision support systems. The ENVRI RIs have started implementing the required technologies at their RI level and

develop first scientific demonstrators for ECV data interoperability of the ATMO-RIs and beyond. Finally, ECV data can be used by interdisciplinary research groups for data assimilation, data fusion, and model development. Some examples from atmospheric sciences are the Copernicus services CAMS and C3S, Earth Digital Twin systems, and the WMO Greenhouse Gas Monitoring Infrastructure, all still under heavy development.

One prime example for research-based information for climate policy is the validation of emission reductions required as part of the COP21 Paris Climate Agreement of 2015. The mitigation measures and the speed of their implementation need to be validated by independent methods and closely monitored, while the influence of natural feedback due to the ongoing climate change will require attention, as this may force a change in the speed of implementation of mitigation measures and adaptation. The contribution of ATMO-RIs to the validation exercise of emission reductions is described by Vermeulen et al. (2020).

**4    Entering the Era of Data-Intensive Science by Integrating Atmospheric Sciences, Data Science and Open Science**

We would like to recall that the research problems behind environmental and societal challenges such as climate change, food security, and natural disasters are intrinsically interdisciplinary. Therefore, the integration of data-intensive sciences and environmental sciences is crucial to improve our skills to create multidisciplinary and cross-domain scientific knowledge which is needed to act on the grand challenges facing our planet and our societies. Modelling these processes individually is

difficult enough, but modelling their interactions is another order of complexity; see, e.g., Bauer et al. (2021) for the discussion of the challenges and opportunities of the digital revolution of Earth-system science, using the example of reliable weather forecast and climate predictions.

Among the indispensable pre-requisites for data interoperability is the organisation of data management systems at each RI in

compliance with the FAIR principles. The entire ENVRI cluster has organised the implementation of the FAIR principles along the "FAIR Pyramid" proposed by Bailo (2019); see Figure 6 for details, particularly with respect to the individual FAIR principles, also shown in this illustration. As can be seen in Figure 6, huge and comprehensive efforts are required to accomplish FAIR scientific data management. Most of the FAIR enabling resources need to be implemented at the level of data and metadata management to comply with the respective FAIR principles. Once this foundation has been set up properly,

the access to science-based services and the use of services can be realised.





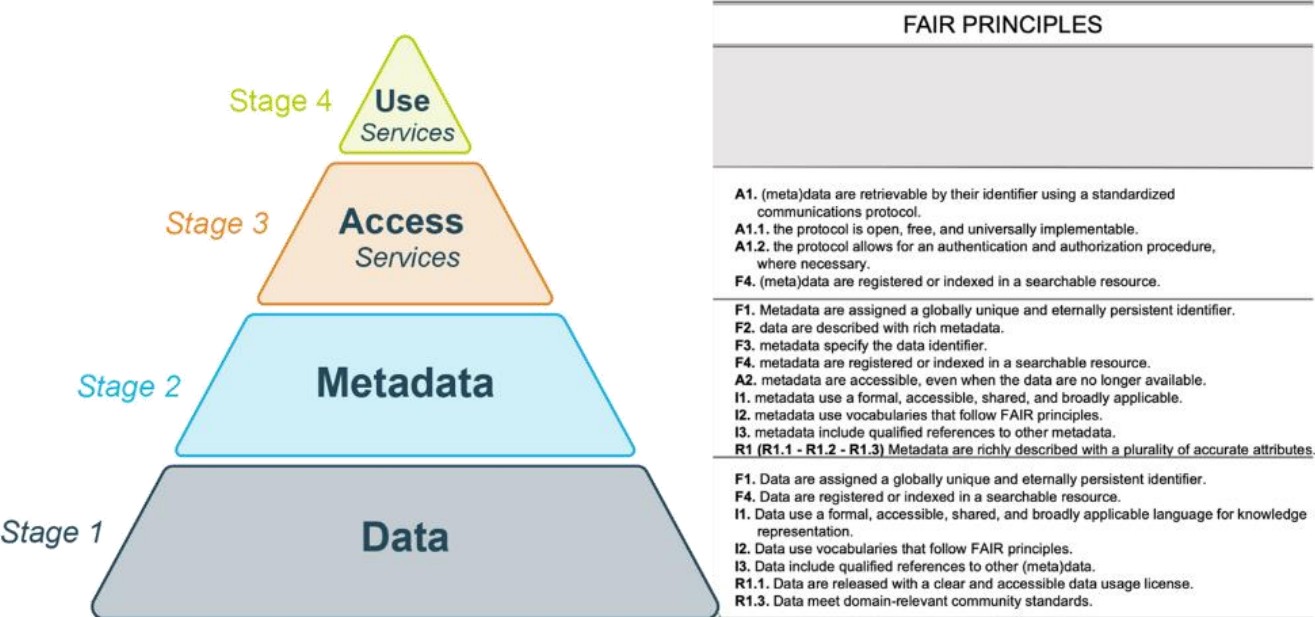

**Figure 6.** The Four-stages FAIR Roadmap - FAIR "Pyramid" (Bailo, 2019).

The RIs of the ENVRI cluster, including the contributing ATMO-RIs, have walked this arduous road during the ENVRI-FAIR project. The reward at the end of the road was a previously unattainable level of data interoperability among the ATMO-RIs which now allows the development and provision of cross-RI scientific services to the scientific community.

In today's atmospheric sciences research area but not only there, scientists are challenged to collaborate across conventional
disciplinary boundaries. However, first they must discover and extract data that is dispersed across many different sources and in many different formats. Effective research support environments are needed for various user-centralised research activities, from formulating research problems to designing experiments, discovering data and services, executing workflows, analysing and finally publishing the results (Zhao and Hellström, 2020). Particularly, the concept of scientific workflows has developed into a powerful tool over the past decade (Atkinson et al., 2017). Scientific workflows provide a systematic way of describing
data analysis with its analytical activities, methods and data needed (Cerezo et al., 2013). They consist of a series of activities with input and output data and are directed to the description of scientific experiments and data analysis processes. Today, data-intensive workflows are capable to exploit rich and diverse data sources in distributed computing platforms and can be composed for re-using elements from other analytical processes in own applications; see also Chapter 5 in Bouwer et al. (2022).



The full research data life cycle for ENVRI RIs shown in Figure 7 illustrates all elements to be contained in such research
support environments, including the publication of data as a critical step for Open Science. Establishing those effective research
support environments is best achieved through a collaborative culture of environmental scientists and computer engineers built
on a co-design approach, and the implementation of mixed teams consisting of team members from both professions
(Schulthess, 2015; zu Castell et al., 2022). Access to FAIR environmental data and research support environments will be
brought to the atmospheric and environmental research communities via the ENVRI-Hub which is described in Section 4.1.

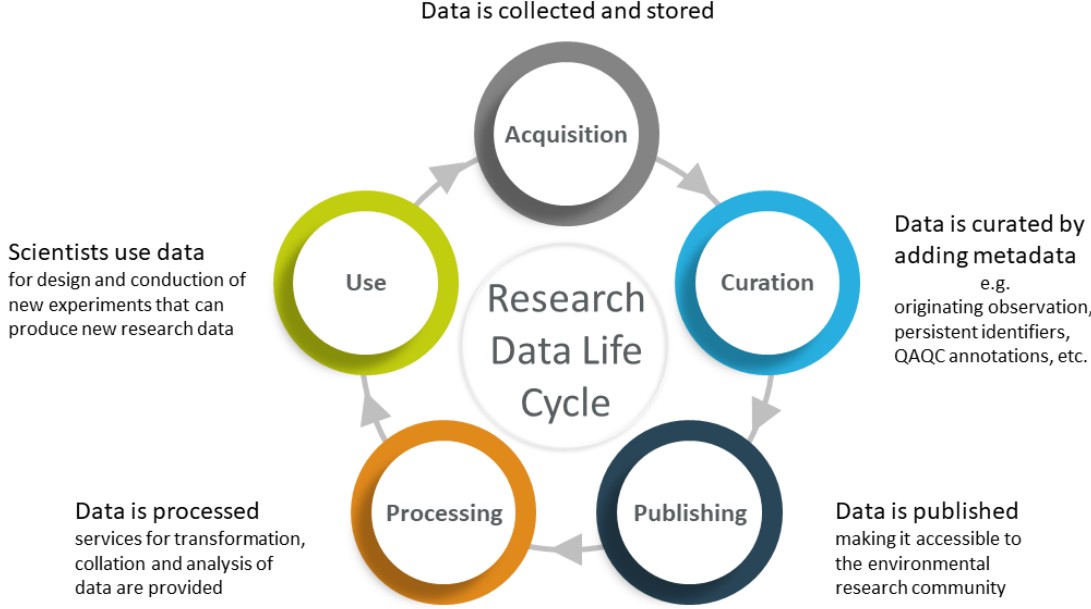

**Figure 7.** The research data life cycle of ENVRI; adapted from Hidalga et al. (2020).

**4.1 The ENVRI-Hub**

The ENVRI cluster responded to the numerous challenges of data-intensive science by establishing a "Data for Science" theme
in their collaborative project ENVRIplus and then by starting the development and implementation of the identified research
support environments in the successor project ENVRI-FAIR. These efforts finally evolved towards the ENVRI-Hub as a
platform for environmental services, data, knowledge, and training to support and increase collaborations and sharing of
information beyond the traditional scientific communities (Petzold et al., 2019; Zhao and Hellström, 2020). Since Open
Science is at the core of the ENVRI RIs, the ENRI-Hub serves also as a robust conceptual and technical framework that will
facilitate the integration of the environmental sciences community into the European Open Science Cloud (EOSC). The
elements of the ENVRI-Hub are sketched in Figure 8. Details of the ENVRI-Hub architecture, its design criteria and
implemented technical solutions can be found in the ENVRI-Hub White Paper (Petzold et al., 2023).





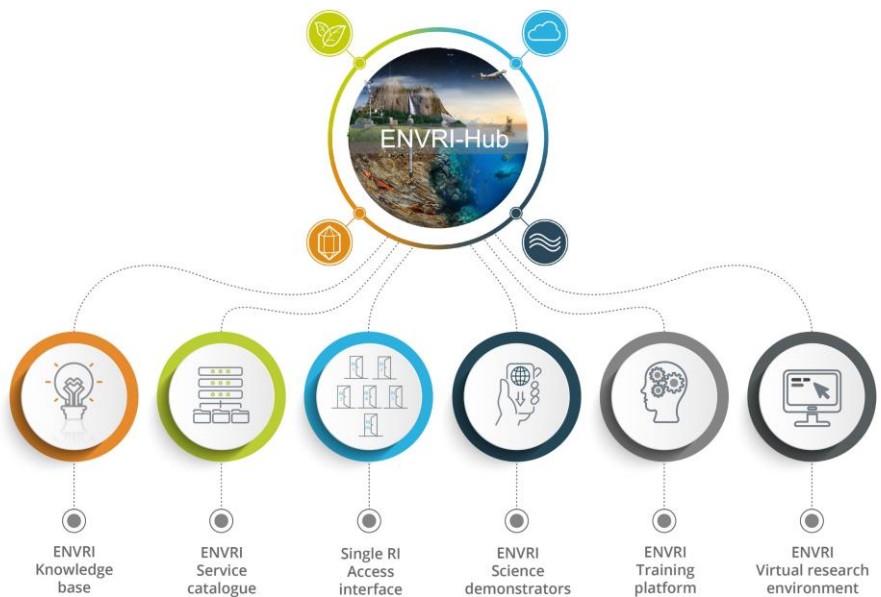

**Figure 8.** The elements of the ENVRI-Hub (Brus and Petzold, 2021).

Currently, the ENVRI-Hub is accessible as a demonstrator version (ENVRI-Hub) with envisaged functionalities not yet fully developed. In its matured stage, the ENVRI-Hub will provide researchers, from e.g., the atmospheric sciences community,

with the necessary services to practise Open Science using data and services published in the ENVRI Service Catalogue, adhering to FAIR principles. It will provide research products and services to EOSC to enhance researchers, citizens, and SMEs involvement in Open Science. ECV workflows, datasets, services, training material and publications will be made available to EOSC when assets and services reach the adequate maturity level. To ensure reproducibility of results, e.g., training and documentation of services, metadata or analytical workflows will be provided and made available through the training

platform assisted by training events. Science demonstrators will be enabled by analytical frameworks, where ECV-based workflow templates will provide the best data handling and visualisation experiences. Uses will include data cleaning and transformation, numerical simulation, statistical modelling, data visualisation, machine learning, and more. Already today, the ENVRI Community in Zenodo (ENVRI-Community) contains documentation, reports and research papers with articles having Open Access licences with minimal restrictions. The release of ENVRI-Hub and the workflows developing on ENVRI-Hub

will be open source managed on the Git repository on EGI infrastructure. The mature releases will also be published in Zenodo.

### 4.2 Science Demonstrators as Examples for Interoperability of Observation Infrastructures

Atmospheric scientists entering the ENVRI-Hub though its portal on the European Open Science Cloud will reach graphical user interfaces for accessing data and services, such as map-based portals and lists of catalogued datasets, with semantic-driven search functionalities that enable them to search for appropriate datasets or services using controlled vocabularies of discipline-



related keywords. They can use the research support environments in different ways, as is illustrated in Figure 9 for the
discovery and use of data (left) or data products and services (right), and build their own analytical workflows from components
developed beforehand by other scientist and published through the ENVRI-Hub.

To give a brief description of data search and discovery (Figure 9, left panel), the scientist browses the metadata catalogue

with the help of the vocabulary, discovers data of potential interest, browses then the metadata catalogue for access information
to the RI of interest with the help of the provided access documentation, and repeats this chain until all data of interest have
been collected. Finally, the whole process can be put into a workflow which can be stored for re-use and composability. A
similar procedure is possible for the advanced service discovery and use (Figure 9, right panel).

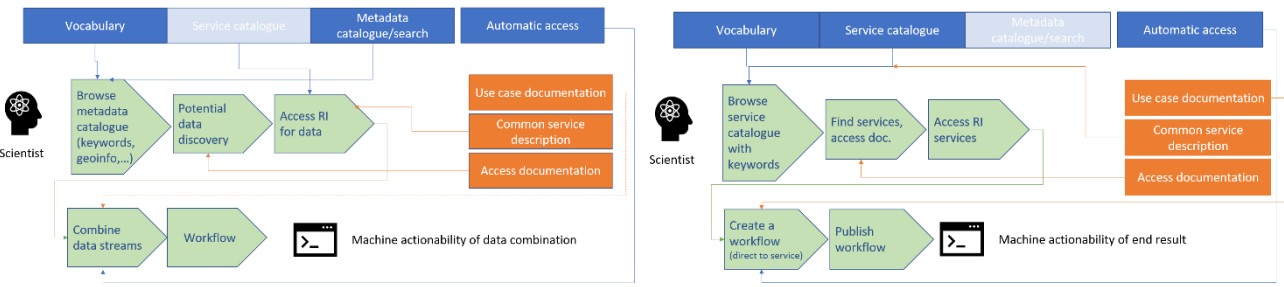


**Figure 9.** Left: Advanced Data Discovery and Use; Right: Advanced Service Discovery and Use with the data in, e.g.,
Jupyter Notebook across RIs (Petzold et al., 2023).


The ground for offering the advanced services illustrated in Figure 9 for atmosphere-centred research questions was prepared
through the considerable improvement of the cross RI atmospheric services and interactions achieved during ENVRI-FAIR.
As a result, it is now possible to implement cross RI services relying on the convergence between RIs on important topics like
vocabularies, unique identifiers, and licenses. The vocabulary within the ATMO-RIs was very diverse, RI specific and

developed over long time from various user communities and applications. Presently, there is a controlled vocabulary
extending beyond our RIs that has enabled better integration with other RIs and thematic domains. The implementation of
unique identifiers was crucial for cross RI synergies and has led to improved exchange of services, procedures, and data
products. Also, an atmospheric community standard has evolved on the use license on data. This led to reliable re-use of data.
With a common use of license now considered more as a "community standard", there is also possibility to produce new high

level cross RI data products utilizing services and data across the Atmosphere domain. The convergence of solutions across
the atmosphere-centred RIs has led to new products useful for all users across the RIs and outside the Atmosphere-domain,



e.g., satellite data extraction and colocation through machine-to-machine interoperable services relevant for the full atmosphere domain and beyond.

Science-based services are designed to facilitate data discoverability and accessibility for complex research questions, and thus to support researchers focusing on scientific questions instead of searching for data. The following examples from the service portfolio of the ATMO-RIs are already accessible through the current demonstrator version of the ENVRI-Hub:

(1) The atmospheric colocation service is a suite of tools to identify and retrieve satellite observations that match ground-based measurements, given a list of stations and satellites and user-specified colocation criteria. The service builds

on the implementation of the ENVRI Python Library for dynamic retrieval of platform metadata for ACTRIS, ICOS, and SIOS stations, and in case of IAGOS for airports for which vertical profiles of atmospheric parameters are provided, and the hierarchical catalogue structure for satellite data built from the AERIS/ICARE information system augmented with specific metadata. The service is accessible via a RESTful application programming interface (API; access point + openAPI documentation), or a graphical user interface (GUI) to build meaningful API queries and

visualize results. Data can be downloaded for direct use in own research applications.

(2) The ICOS Carbon Portal Footprint tool provides atmospheric surface emission influence calculations (a.k.a., footprints) for any 3D location or specific station in the model domain and any period for which meteorological data are available. When ready, the results can be viewed in the tool and directly accessed through our Jupyter notebook service and Python library, to be combined with emission data for concentration calculations at the receptor(s) and

comparison with observations retrieved from the RI data repositories.

(3) The FAIR ENVRI Atmospheric Data Demonstrator aims to shorten time response in case of extreme events like fires, emissions form volcanic eruption and desert dust intrusion by providing scientific analyses as well as harmonized datasets and tools. It will offer the search for the ATMO-RIs data availability for the period and area of interest with a focus on ECVs, fetch the selected data including automatic previews of the datasets and provide statistical analysis

on the downloaded time series. In addition, access to data from ATMO-RIs is offered as a bundle with automated compilation of provenance information, and a co-location service with satellite data is available.

(4) The Dashboard for the State of the Environment is designed to be completely user configurable so that the users can select from a list the indicators to be shown and their order. Providers can add, remove, and edit indicators through a standard RESTful API, that allows transferring all parameters, including the configuration of the indicators and how

to provision data values and thumbnail interaction. The Dashboard is implemented and operated using engineering best practices and a robust and flexible container-based deployment. It builds on EOSC services like AAI, cloud services, and data storage, and the workflows that provide the indicators will also build on the EOSC and Research Infrastructure (RI) computing integration. As a proof of concept, a limited list of indicators is currently available, and we foresee that the participating RIs will provide many more indicator options in the near future. The Dashboard

service is completely open source, and, as the whole concept, it is designed to be flexible and expansible.



The science demonstrators are practical examples for the provision of interoperable services from the ATMO-RIs to the scientific community, enabling cross-RI data discovery and use. The software behind the science demonstrators can be accessed in github repositories for further use and composition with own workflows. In that respect, the science demonstrators are showcases for new research opportunities evolving from the combination of RI data infrastructures with tools and methods

of data-driven science. In the following section we will expand the specific examples to a broader description of new tools and methods for scientific knowledge production.

### 4.3 New technology approaches supporting scientific knowledge production

Currently, considerable knowledge is recorded in scientific articles. This narrative text document-based expression of research work is designed for human expert consumption; for machines, knowledge recorded in scientific articles is not accessible. As

a result, machine support in scientific knowledge processing is inadequate and research communities routinely invest enormous human resources to conduct research synthesis, e.g., in systematic reviews, as information needs to be manually extracted from articles and manually organized to support synthesis. In the coming decade, scientific knowledge infrastructures (Stocker, 2017) must further advance their digital infrastructures so that scientific knowledge is produced FAIR by design, therefore more efficiently reusable with advanced machine support. Initiatives and services such as the Open Research Knowledge

Graph (Stocker et al., 2023), are demonstrating how advanced technology such as Knowledge Graphs, Natural Language Processing, Terminology Services and Semantic Resources, and approaches that integrate the production of FAIR scientific knowledge into data analysis and that enable the reuse of FAIR scientific knowledge in data science, can fundamentally transform digital scholarship and the way digital infrastructures support scientific knowledge production.

Knowledge Graph, and, more generally, Artificial Intelligence technologies, Virtual Research Environment (VRE) services, and FAIR scientific data management are key instruments supporting the production of new scientific knowledge. Data discovery services, such as the ENVRI Knowledge Base search engine (Farshidi et al., 2021), can improve the search quality of research assets by 1) allowing search services to effectively expand search queries with a broader set of relevant and similar keywords and 2) enhancing the similarity inferences of search results (e.g., keyword indexing-based methods in classical

information retrieval), using extra semantic information provided by the concepts and their relations in the Knowledge Graph (Farshidi and Zhao, 2022). Using semantic information provided by Knowledge Graphs, a scientific experiment or observation strategy can be annotated with rich meta information for describing its domain problem, data sets, software tools and experiment steps, which allows for effective sharing and reuse of experiment workflows among user communities. As new knowledge is derived from scientific experiments or observations, the publications of the scientific paper, research software

and data sets will be further linked in the Knowledge Graph.



## 5    Opening up New Research Opportunities

### 5.1    Re-thinking the scientific approach

Performing atmospheric research through the scientific data and services provided by the ATMO-RIs and making use of the opportunities offered by Open Science requires a fundamental change in the mindset of the individual scientists but will open

up new ways of tackling scientific questions of increasing complexity. Particularly, adaptation to the approaches and methods of data-intensive science applied to open data, and services from long-term operating research infrastructures calls for rethinking at multiple levels:

On the level of data provision, the creation, curation, and processing of observational data is disconnected from data analysis

and scientific exploration, and data provision is realised by the ATMO-RIs through the ENVRI-Hub or other open-access data sources, supported by rich metadata. On the level of scientific exploration, large data sets will remain at the source while data analysis is brought to the data by means of science-based composable workflows, running in analytical frameworks such as Jupyter notebooks. Open-source programming environments like github foster the collaborative development and application of data-analytical tools and methods in science teams which are ideally composed of both domain scientists and software

developers and are restricted neither institutionally nor regionally because of the technologies for remote work offered by the worldwide web.

On the conceptual side, the implementation of the science-based framework of Essential Climate Variables for the provision of long-term and high-quality data by the ATMO-RIs for atmospheric research and beyond, combined with the transparent

reporting of data handling and processing in accordance with the FAIR principles, i.e., provenance information as part of rich metadata, and the discoverability of data via enhanced search capabilities though standardised keywords and vocabularies, significantly broadens the database accessible to scientists. Making use of open-source programming environments linked with harmonised standard operating procedures for instruments measuring ECVs in accordance with the GCOS defined requirements for data accuracy and precision, promotes the collaborative development of data quality assurance and control

procedures across the ATMO-RIs, culminating in an increasingly consistent dataset of ECVs. In analytical frameworks made available by the ATMO-RIs as well as other ENVRI-RIs through the ENVRI-Hub, ECV-based scientific workflow templates will provide the best data handling, processing, analysis, and visualisation experience. Uses include data cleaning and transformation, numerical simulation, statistical modelling, data visualisation, machine learning, and more.

New ways of scientific knowledge creation are opening up when leaving the traditional path of working with self-generated data or data from a scientific collaboration and analysing the data within the involved research teams. Recalling that data-intensive science is characterised as collaborative, networked, and driven by immense amounts of data, the access to open data, open software, and resources of previously unknown computational power, combined with a positive attitude towards





knowledge-sharing indeed offers a new scientific approach towards complex scientific questions which was not possible
before. However, one of the most important bottlenecks in this context is the training of current and future scientists working
in our field to familiarize them with the new concepts, tools, and methods.

## 5.2     Research Infrastructures as key facilitators of the new scientific approach

The environmental research infrastructures organised in the ENVRI cluster have covered a considerable part of the way
towards a harmonised global climate observation system complemented by open access and interoperable reliable data, as
requested by WMO (Weatherhead et al., 2018; Carmichael et al., 2023), or by the initiative taken by Kulmala and co-workers
(2018; 2023a; 2023b). Making data and scientific products available through free and open access, further broadens the use
and re-use of the scientific information provided by ACTRIS, IAGOS and ICOS, and finally the scientific knowledge gained
from the continuous operation of the ATMO-RIs. Already today, the atmosphere-centred environmental research
infrastructures play a central role in the evolving global climate observing system through the provision of scientific data and
knowledge to institutional users like the Copernicus Atmosphere Monitoring Service or the Global Atmosphere Watch
programme of the World Meteorological Organisation. Through the efforts undertaken by the ATMO-RIs jointly with the
other RIs of the ENVRI cluster towards the provision of FAIR data by open access and supported by analytical frameworks,
the benefits for the global scientific community in climate sciences and beyond will continuously increase and offer new
research options, particularly to the new generation of scientists familiar will data-intensive tools and methods


Continuing on the outlined path to FAIR and open data in the environmental sciences will not only open the door to broader
use of data and scientific products by the global scientific community, especially across traditional scientific fields, but will
also foster the deeper integration of environmental, life, and social sciences, needed to respond to the grand societal challenges
we face. The emerging facilities offered by the European Open Science Cloud and the ENVRI-Hub as EOSC-hosted platform
for collaboration and co-design of the ENVRI RIs will play a central role in this integration of research infrastructures towards
interoperability of data and services.

It must be noted, though, that during this integration process the integrity and functionality of the independent research
infrastructures need to be maintained. Their operational requirements are very different and call for tailored solutions for
infrastructure design and implementation. Integration takes then place at the level of providing scientific data and services, as
well as access to the individual infrastructures. The sustainable success model therefore consists of highly specialized, unique,
and independent research infrastructures, that integrate as much as possible in the provision and interoperability of data,
services, and access to research infrastructure digital and physical assets of any kind.




**Author contributions.**

AP conceived and wrote the manuscript, and all co-authors contributed to their respective areas of expertise.

**Competing interests.**

At least one of the (co-)authors is a member of the editorial board of Atmospheric Chemistry and Physics.

**Acknowledgements**

The European research infrastructures ACTRIS, IAGOS and ICOS gratefully acknowledge the continuous support by the European Strategy Forum on Research Infrastructures, ESFRI, and the long-term funding by the European Union's Research and Innovation programmes through the cluster projects ENVRI (2011-2014; Grant Agreement 283465), ENVRIplus (2015-2019; Grant Agreement 654182), ENVRI-FAIR (2019-2023; Grant Agreement 824068), ATMO-ACCESS (started 2021, Grant Agreement 101008004), RI-URBANS (started 2021, Grant Agreement 101036245), ICOS Cities (started 2021, Grant Agreement 101037319), and EOSC Future (started 2021, Grant Agreement 101017536). Furthermore, all RIs gratefully acknowledge the support by the European Union's Research and Innovation programmes during the design and implementation phases of the individual RIs, and the support of the operation by all involved national research ministries and funding agencies. Finally, the authors thank the Executive and Senior Editors of Atmospheric Chemistry and Physics for the opportunity to contribute this Opinion Article to the Special Issue for 20 years of Atmospheric Chemistry and Physics.

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
