# Peer review of "Opinion: New directions in atmospheric research offered by research infrastructures combined with open and data-intensive science"

_EGUsphere, 2023_

## Author Comment (AC1)

**Reviewer #1**

The authors are grateful to Oksana Tarasova for her very positive and helpful comments which definitely helped to sharpen the article and make the important messages more visible. Below, we respond to Oksana's comments point by point.

The paper "New directions in atmospheric research offered by research infrastructures combined with open and data-intensive science" provides useful view on the future of the atmospheric research. The paper contains a lot of information and in its current form it rather looks like a comprehensive review than an opinion. It will take a reader 25 pages of reading until the authors go to the point.

I would suggest that the author shorten the paper by at least 30% , which still would allow to make a point about future research approaches. In particular, this applies to the references and description of the infrastructures where the whole paragraphs are just copied from the referred publications.

**Reply:** To make the key messages more visible, we moved the entire former Section 3 on Research Infrastructures into an Appendix. However, we do not delete this section completely since from our discussions at multiple dissemination events in our research communities, we still recognized that not many scientists and in particular young researchers are aware of what research infrastructures are and how they can make use of them.

By rearranging the manuscript, the key message appears now much earlier, but the background information on Research Infrastructures is still available for those who are not familiar with this new tool.

I find the authors opinion interesting, though it would be useful to have authors opinion on two points which are currently not reflected in section 5. The first one: currently the performance of scientists is judged by a number of publications they produce. Within the envisioned new research environment, the research will be done as a collective one. This diminishes the role of the individual researcher and the current evaluation framework by publication number (as in the envisioned research environment the summaries can be easily generated by AI). The second question is related to the role of innovation (e.g. regarding analytical methods or measurement techniques and instruments) within the highly standardized research environment.

It would be useful if the authors reflect on these points in the next revision of the paper.

**Reply 1/:** The need to revise the reward system for researchers is being discussed as part of the open data and open source movements. In response to this discussion, the citation of datasets and their link to the responsible researchers has been improved by adding DOIs to datasets and referencing them in publications. This cultural change is taking place across the scientific community. It is also called for in the European Open Science Cloud Declaration of 2017 (https://eosc-portal.eu/sites/default/files/eosc_declaration.pdf). For this reason, we have decided not to focus on this issue as it is independent of the atmospheric sciences.

**Reply 2/:** The role of innovation, e.g., regarding analytical methods or measurement techniques and instruments, within the highly standardized research environment cannot be underestimated. Research infrastructures not only operate instruments following standardized operation procedures but improve existing or develop new techniques as part of their mission. This is an intrinsic activity of the research infrastructures. However, this activity is independent of the combination of data intensive science and research infrastructures, which is the topic of this opinion article. For this reason, we decided to leave the topic of technological innovation untouched in our opinion article.

General note on the use of language: the sentences are too long, and the idea is often get lost before one reads to the end. It would be magnificent if authors use more concise language.

**Reply:** The entire manuscript has been revised for a more concise language.

**Additional comments:**

**1.** Line 22: the term "variability" may be a bit better than "periodicity"

**Reply:** Thank you for this suggestion, we use now seasonality instead of periodicity.

**2.** Line 31: "which evolve from"; it is not clear from what opportunities evolve as you do not provide initial state of affairs. Maybe it would be better to use another works here, like "emerge"

**Reply:** Thank you, we use now "arise".

**3.** Line 32: what do you mean with "emerging service ecosystem"

**Reply:** We used more concise language and write now: "and the developing collaboration platform ENVRI-Hub, hosted by the European Open Science Cloud."

**4.** Line 44: "economic processes in the Earth system" – economic processes refer to human society rather than to the Earth system

**Reply:** We replaced "Earth system" by "on our planet" which includes both the Eart system and human activities.

**5.** Line 50: GCOS is a co-sponsored programe of WMO, IOC-UNESCO, UNEP and International Science Council

**Reply:** Thank you, this remark is included.

**6.** Line 55-56: do you consider vertical profile measurement as ground based or as in situ?

**Reply:** We specified as "ground-based and airborne in-situ observations".

**7.** Line 62-66: lots of repetitions here

**Reply:** This paragraph has been significantly shortened.

**8.** Line 67: actually the threats are posed to humanity rather than to the planet

**Reply:** The threats of climate change target the human society but also biodiversity in all compartments of the Earth system. Therefore, we prefer to keep "our planet".

**9.** Line 94-94: I would disagree with the comparison with astronomy infrastructure. Unlike astronomy, environmental infrastructure has an immediate value for multiple applications, including climate services for mitigation and adaptation, health and agriculture applications, hence this poses different requirements for timeliness of data availability.

**Reply:** The comparison with astronomy refers to the implementation of tools for data intensive science. However, the statement on the timeliness of data availability is highly valuable and added to the paragraph. The full paragraph reads now:

"Astronomy and high-energy physics can serve as role models for this kind of infrastructures (Bell et al., 2009), but with the essential difference that environmental observations always require the

operation of largely distributed measurement networks together with the handling of complex data streams from highly diverse sources and the need for short-term data availability."

**10.** Line 120-126: it is not clear what point you are trying to make in this paragraph

**Reply:** Infrastructures and networks like ICOS contributed significantly to the data base behind the Nobel Prize 2021. Therefore, we want to keep this paragraph.

**11.** Line 146-151: not clear what the relevance of this paragraph to this paper

**Reply:** This paragraph was removed for the sake of conciseness.

**12.** Line 154: air quality is a part of atmospheric composition

**Reply:** This is correct, but we wanted to mention it explicitly. So, we added "atmospheric composition in general".

**13.** Line 170-176: another paragraph of repetitions

**Reply:** This paragraph was removed for the sake of conciseness.

**14.** Line 187-190: the name of infrastructures should be spelled out here, rather than later in the text (lines 205-210)

**Reply:** This paragraph was removed for the sake of conciseness. Details refer only to the atmosphere-centric infrastructures.

**15.** Line 189: I was under impression that ACTRIS also recently became ERIC

**Reply:** That is correct, the reference is updated.

**16.** Line 203: it is not necessary to spell out the title of the book, reference would be sufficient

**Reply:** The title of the book was removed.

**17.** Line 214: "in providing" ("in" is missing)

**Reply:** Corrected.

**18.** Figure one: the name of the phenomena is "sand and dust storms" not "desert storms"

**Reply:** Corrected.

**19.** Table one: "aerosol particles" – use either of the words, not both

**Reply:** From a scientific perspective, the term is correct since it refers to the particle phase of the aerosol which is composed of both the particle and the gas phase.

**20.** Comma is missing after O3 in IAGOS section (Table 1)

**Reply:** Corrected.

**21.** Line 237: what do you mean with "research infrastructure process"?

**Reply:** We added "implementation process" for clarification.

**22.** Line 240-241: "Before the consolidation phase, data have been locked in silos and were poorly standardised over decades, hampering scientific progress severely" – I disagree with this statement as it diminishes the role of the programmes like the Global Atmosphere Watch

**Reply:** We modified the statement and included the role of GAW. The sentence is now: "On a global scale, first harmonisation efforts were undertaken by the Global Atmosphere Watch programme of WMO. During the consolidation process, the ATMO-RIs have developed cutting-edge standardisation for their observations and implemented highly efficient data life cycle management systems, leading to timely, open, and FAIR data access by scientists all over the world."

**23.** Line 251: "periodicity in the presence of" is better to reformulate as "variability in atmospheric levels/burdens of"

**Reply:** as suggested by Reviewer #2, we rephrased the sentence as follows: "The detection of trends and seasonality in the presence of greenhouse gases … ".

**24.** Line 259-296: what are you trying to demonstrate with the presented three examples?

**Reply:** Among the most important achievements of Ris is the securing of long time series collected in projects which would have been lost otherwise after the finalization of these projects. This is what we want to demonstrate with the presented examples.

**25.** Line 277: "Data which were collected before ICOS measurement protocols were put in place, have been secured physically". Could you please explain this statement? Most of ICOS atmospheric data are part of GAW and those data were and are achieved in the World Data Center for Greenhouse Gases supported by Japan

**Reply:** This statement refers to the work performed by the RIs for lifting the historic greenhouse gas data to the level of the ICOS data protocol. In this respect, there is an additional achievement of ICOS beyond archiving station data in the GAW data base.

**26.** Line 286: the referred "latest assessment" is 5 years old. Are there updates on this publication?

**Reply:** The assessments are published every 5 years. So, a new assessment is in preparation but not yet published.

**27.** Line 299-300: could you please clarify the statement "Trends are detectable in the data, possible only due to collection of data with consistent quality and operating procedures over long time periods."

**Reply:** Trend detection requires long time series of consistent data quality. We believe that the Ris make a significant contribution to the provision of time series which can be used for trend analyses.

**28.** Line 319: ICOS is a regional network

**Reply:** Of course, the ICOS observation network is densest in Europe, and in that sense it is of continental scale. But still, ICOS is also the densest greenhouse gas observation network worldwide. Therefore, we suggest to keep the formulation.

**29.** Line 322: "which investigates the processes" – who investigate the processes?

**Reply:** We refer to ACTRIS as an infrastructure which is investigating the atmospheric processes.

**30.** Line 323: what do you meant with "controlled atmospheres"?

**Reply:** The term "controlled atmosphere" refers to the concept, that we can fill atmospheric simulation chambers with an atmosphere of known composition and investigate chemical processes under these specific conditions.

**31.** Figure 4: I disagree with the proposed adjustment of the figure as it completely changes its sense. The original figure was designed to resent a limited set of key climate indicators and each box includes only one variable, while the added box includes tens of variables disturbing the intended meaning. I would suggest to remove this figure from the paper.

**Reply:** In accordance with the response to a comment by Reviewer #2, we decided to use the original figure here. The sentence describing this figure has been adapted.

To make the difference between GCIs and ECV – which include short-lived climate forcers – clearer, we rephrased this paragraph to:

"To transform the challenging task of integrated Earth observation into a concept towards a global climate observation system, WMO/GCOS has defined a set of global climate indicators (WMO-GCI), which stretches beyond the boundaries of traditional scientific disciplines like, e.g., atmospheric sciences, ocean sciences, or biology; see **Fehler! Verweisquelle konnte nicht gefunden werden.**. This set of GCIs defines the indicators which point to long-term changes in the Earth system. Consequently, WMO requests the continuous observation of these indicators for monitoring the state of Earth's climate. However, this undertaking requires a largely interdisciplinary approach.

To further develop this interdisciplinary approach towards a global climate observation system, WMO has introduced the concept of essential climate variables supplementary to the GCIs. Essential climate variables for atmospheric composition are defined as a physical, or chemical variable or a group of linked variables that critically contributes to the characterization of the atmospheric composition. This set of variables includes the long-living greenhouse gas $CO_2$, but also many of the short-lived climate forcers."

**31.** Line 346: "this approach" – which one?

**Reply:** We refer here to the largely interdisciplinary approach needed to build such a global climate observation system. This approach is mentioned in the previous sentence, but for clarification we added the term "interdisciplinary". See also reply to item 31.

**33.** Line 348: could you please provide the source for the definition of "essential variable". Is it introduced by the authors of the paper?

**Reply:** We used "essential variables" as synonym for essential climate variables. For clarification, we rephrased the sentence. See the reply to item 31 for the modified paragraph.

**34.** Line 353: where does this number of 12 application areas come from. There are 10 application areas under atmospheric domain as one can see in the OSCAR database https://space.oscar.wmo.int/applicationareas

**Reply:** Thank you for pointing at this error. We updated the number.

**35.** Line 379: OSCAR is a database, not a table.

**Reply:** To stay consistent with the WMO nomenclature, we replaced "table" by "tool".

**36.** Table 2a: is the word "distribution" missing in the "aerosol number and size"? In GAW most of the ozone depleting substances are included under greenhouse gases

**Reply:** Thank you, the term "distribution" was added.

**37.** Table 2b: GCOS does none use volume mixing ratio for ozone variables. The correct variable is mole fraction

**Reply:** Thank you, the unit was adjusted.

**38.** Line 405: please use either upper air or upper atmosphere, not both

**Reply:** This sentence was removed during the reorganization of the text.

**39.** Figure 5: typo at the end of the first line ("to;" is not needed)

**Reply:** Figure caption was corrected.

**40.** Line 419: I guess "hundreds of stations" applies to temperature observations as this is not the case for greenhouse gases

**Reply:** The word is just used as a synonym for "many". We replaced the term by "numerous".

**41.** Line 430: the term "validation" is not used in the Paris agreement

**Reply:** The term "validated" was replaced by "checked".

**42.** Line 444: what do you mean with "organised the implementation of the FAIR principles". The principles can be applied to something, rather implemented

**Reply:** For clarification we replaced "implementation of the FAIR principles" by "implementation of the respective FAIR enabling resources".

**43.** Line 448: "FAIR enabling resources need to be implemented" – resources can be used, rather than implemented

**Reply:** In this context the phrase "implementation of FAIR enabling resources" is frequently used. So, we prefer to keep this sentence.

**44.** Line 459-468: is this paragraph needed? Line 468: "composed of" or use another word instead of "composed"

**Reply:** The content of this paragraph is needed because composable workflows are a key tool of this new scientific approach. The paragraph was shortened and significantly rephrased.

**45.** Line 487: how satellite and modelling data re integrated in this cloud?

**Reply:** The integration of data and services into the EOSC is driven by the providers of these data. Thus, we can speak only for our involved Research Infrastructures.

**46.** Line 496: could you please address the risks of the data misuse and misinterpretation by citizens without appropriate background? Training platforms cannot substitute years of professional education.

**Reply:** The issue of misuse or misinterpretation of data in citizen science is a huge one, and certainly beyond the scope of this article. As citizens are only one of many users of environmental data, we prefer to refrain from commenting on this topic. This would require a separate article.

**47.** Line 497: please spell out SMEs

**Reply:** "SMEs" is replaced by the general term "industry".

**48.** Line 501: could you please use another word for "uses"

**Reply:** "uses" was replaced by "applications".

**49.** Line 505: please spell out abbreviations here, it reads like a lot of slang

**Reply:** The terms GIT repository, EGI and ZENODO are not acronyms, but are used to refer to open source programming tools (GIT), a large European e-Infrastructure (EGI Foundation) and the Open Access repository operated by OpenAIRE.

**50.** Line 515: "metadata catalogue to access information from the RI of interest"

**Reply:** WE clarified the phrase as: "browses then the metadata catalogue for information about the access to the RI of interest"

**51.** Line 517: "composability" -please clarify

**Reply:** The composability of workflows is a standing term in the context of workflows. For clarity, we rephrased it as "for re-use and combination with other workflows".

**52.** Line 533: "on the use of license on data"

Reply: The sentence was rephrased as : "Also, an atmospheric community standard has evolved on the license for data use".

**53.** Line 578-579: "for machines, knowledge recorded in scientific articles is not accessible" – this is incorrect and the text below referring to the AI had been used for this purpose. Later in the text (line 599) the authors refer to the publication of the scientific papers, which according to the initial statement would be a waste of resources

**Reply:** The point we want to make here is that knowledge recorded in scientific articles on a narrative basis is indeed not accessible to machines without the help of humans who identify relevant knowledge in publications and transform it into a machine-readable format. This cannot be done by AI. However, the publication of scientific papers will always be one way of generating and disseminating scientific knowledge, but there will develop other ways, too. Therefore, we prefer to keep this paragraph as it is.

---

## Author Comment (AC2)

**Reviewer #2**

This manuscript offers an opinion on "new directions in atmospheric research offered by research infrastructures (RIs) combined with open and data-intensive science". The European author group is well qualified to discuss the present and future of research infrastructures based on their involvement and experience. Recognizing the current state of affairs of observations, data and models related to climate resilience and anticipating future needs and demands on the RIs of the world is the basis of a valuable narrative. The manuscript is suitable for publication after the authors consider the following comments.

**1)** The manuscript would be improved by substantially shortening its length. It is a slow read given the abundance of detail that is primarily background material to the main thesis. In shortening, I suggest a focus on the promised 'new directions', ie to make them standout better in this narrative.

**Reply:** To make the key messages more visible, we moved the entire former Section 3 on Research Infrastructures into an Appendix. However, we do not delete this section completely since from our discussions at multiple dissemination events in our research communities, we still recognized that not many scientists and in particular young researchers are aware of what research infrastructures are and how they can make use of them.

By rearranging the manuscript, the key message appears now much earlier, but the background information on Research Infrastructures is still available for those who are not familiar with this new tool.

**2)** The figures (except 6 and 7) and Table 2 seem to be unnecessary information for this opinion piece; hence I suggest the authors considering deleting.

**Reply:** See reply to Comment 1. The illustrations and tables referred to have been moved to the Appendix.

**3)** The most interesting and relevant text was in sections 4 and 5 where the reader finds the most new thinking about this topic.

**Reply:** Former Sections 4 and 5 are ow Sections 3 and 4, to give them more weight.

**4)** The recommendation that the GCIs be modified to include SLCFs appears rather casually on ln 337. The GCIs are formally put forth by WMO/GCOS so a change would be a major consideration. While a quite reasonable suggestion, this proposed change could alone be the topic of an opinion piece. In the present context, it is not clear what the authors would like the reader to think about this proposal, eg what is the next step in advancing this idea or is it only meant to be a marginal comment?

**Reply:** Given the formal nature of the set of GCIs and the consequent major consideration to changing this figure, we agree to use the original figure here. The sentence describing this figure has been adapted. The reason for adding short-lived climate forcers here was that reducing the changing atmosphere to its $CO_2$ content appears to be a simplification and neglects all the processes triggered by short-lived climate forcers.

To make the difference between GCIs and ECV – which include short-lived climate forcers – clearer, we rephrased this paragraph to:

"To transform the challenging task of integrated Earth observation into a concept towards a global climate observation system, WMO/GCOS has defined a set of global climate indicators (WMO-GCI), which stretches beyond the boundaries of traditional scientific disciplines like, e.g., atmospheric sciences, ocean sciences, or biology; see **Fehler! Verweisquelle konnte nicht gefunden werden.**. This set of GCIs defines the indicators which point to long-term changes in the Earth system. Consequently, WMO requests the continuous observation of these indicators for monitoring the state of Earth's climate. However, this undertaking requires a largely interdisciplinary approach.

To further develop this interdisciplinary approach towards a global climate observation system, WMO has introduced the concept of essential climate variables supplementary to the GCIs. Essential climate variables for atmospheric composition are defined as a physical, or chemical variable or a group of linked variables that critically contributes to the characterization of the atmospheric composition. This set of variables includes the long-living greenhouse gas $CO_2$, but also many of the short-lived climate forcers."

**5)** Another casual remark is on ln 430 : 'One prime example for research-based information for climate policy is the validation of emission reductions required as part of the COP21 Paris Climate Agreement of 2015.' The failure to achieve GHG emissions and concentration reductions in the next 2 decades is perhaps the greatest threat to the future health of human society and ecosystems. The role of RIs is essential to have an efficient, effective and verifiable global emissions reductions. I could easily see an opinion piece focused on this essential RI role under the label of New Directions and hence standout more in this opinion piece.

**Reply:** We appreciate this suggestion and implemented it in the manuscript. In current section 2.2 on the impact of cross-disciplinary research supported by ATMO-Ris, we supplemented the paragraph on the role of CO2 observations for the 2021 Physics Nobel Prize by the statement on the relevance of RI long-term observations of the validation of $CO_2$ emissions. Now, this important role of RI contributions appears at a prominent place of the article. The full text reads now:

"The scientific importance of long-term greenhouse gas observations as conducted since many years by the ENVRI RI ICOS, and since recently, by IAGOS, has been confirmed in the Scientific Background document on the 2021 Physics Nobel Prize which went to Klaus Hasselmann, Syukuro Wanabe and Giorgio Parisi "For ground breaking contributions to our understanding of complex physical systems". In Chapter IV, the Nobel Committee writes: "From the perspective of laboratory science, using experimental measurements to test theory is such a self-evident step in the scientific method that it goes without saying. However physical cosmology and physical climatology are observational sciences – practitioners observe that which nature allows."

Moreover, failure to reduce greenhouse gas emissions and concentrations over the next two decades is perhaps the greatest threat to the future health of human society and ecosystems. The role of the ATMO-RI is critical to achieving efficient, effective and verifiable global emission reductions, particularly for the validation of emission reductions required as part of the COP21 Paris Climate Agreement of 2015. The mitigation measures and the speed of their implementation need to be checked by independent methods and closely monitored, while the influence of natural feedback due to the ongoing climate change will require attention, as this may force a change in the speed of implementation of mitigation measures and adaptation. The contribution of ATMO-RIs to the validation exercise of emission reductions is described by Vermeulen et al. (2020)."

**6)** Para at ln 249. It is important to note that using long timeseries of observations for trend analysis requires a measurement infrastructure that guarantees intercomparability regarding observation precision and accuracy over the time period of interest. Such intercomparability generally requires constant vigilance and support.

**Reply:** Thank you very much for this suggestion which we implemented. The text covering this topic is now in the appendix. The full paragraph reads now:

"Another major achievement of the ATMO-RIs which deserves explicit mentioning is the conservation of long-term atmospheric observation stations and time series which were in danger during the transition process from research projects to research infrastructures. The detection of trends and seasonality in the presence of greenhouse gases and short-lived climate-active atmospheric constituents is an important aspect of climate science. An accurate description of trends relies heavily on the ability to place the measurements of all kinds of climate-active atmospheric constituents in a historical context, i.e., to compare measurements against measurements from the same location in preceding years and decades. From a scientific perspective, longer timeseries reduce the uncertainties in the interpretation of current measurements. The length of timeseries which are produced by the different ATMO-RIs is therefore a good indicator of how well these data meet the requirements of climate scientists. It is important to note that the use of long time series of observations for trend analysis requires a measurement infrastructure that guarantees inter-comparability in terms of observation precision and accuracy over the period of interest. Such inter-comparability generally requires constant vigilance and support which operational RIs secure."

**7)** Minor point: suggest replacing 'periodicity' with 'seasonality' since the former is non-standard usage.

**Reply:** Thank you for this suggestion, we use now seasonality instead of periodicity.

---

## Author Response (AR2)

**General remarks on the 2nd reviews of the manuscript egusphere-2023-1423**

The authors are grateful to both referees for their very positive and helpful comments on the revised manuscript. In response to the main criticism of the first version of the manuscript about its length and structure, we have moved the description of research infrastructures to the appendix and sharpened the main text about new research opportunities. This re-structuring is supported by Referee #1, whereas Referee #2 still suggests removing the appendix without significant loss but focus on a to-the-point narrative. Since no reference document exists which summarises the structure and benefits of atmospheric research infrastructures, we prefer to keep the appendix but in an even shorter version.

The following editorial changes were made to the manuscript:

1. The text was again checked for redundancy and concise language.
2. Figure 5 was reproduced with better quality and the colour scheme was adapted to the used one.
3. Table 2 was simplified.
4. One reference was removed, and four references were added on request by co-authors and one reviewer, see the annotated reference list.

In addition to this general change to the manuscript, we are responding to the referees' comments point by point.

**Referee #1**

The paper has substantially improved after review and key messages are very clear. Moving the description of infrastructures and examples in the Appendix and shortening of the text was also very helpful and improved the paper.

Nevertheless, I still have a problem with the section 5.2 that contains factual mistakes. I would strongly advise the authors consulting documentation on the Rolling Review of Requirements at https://space.oscar.wmo.int/applicationareas. One of the applications the authors refer to does not exist, hence this point needs revision.

Another issue with the section 5.2 is misrepresentation of the "global climate indicators". These indicators do not exist. GCOS initially set up a framework of Essential Climate Variables. Then to simplify communication the WMO Commission on Climatology established a set of seven headline climate indicators. The process and reasoning behind the selection is described in Trewin et al., 2021, available at https://journals.ametsoc.org/view/journals/bams/102/1/BAMS-D-19-0196.1.xml

Hence, before publication, section 5.2 needs to be re-written.

**Reply:** All references to global climate indicators and essential variables have been removed from the manuscript, including Figure 7. The revised text concentrates exclusively on essential climate variables. Please check the whole revised section 5.2 in the annotated manuscript.

**Editorial corrections**

There is a number of the small editorial corrections that are needed:

l.23: trends of what?

**Reply:** The full sentence is "In particular, the detection of trends and seasonality in the presence of greenhouse gases and short-lived climate-active atmospheric constituents is an important aspect of climate science." We have replaced "presence" with "abundance".

l.26: please use "discuss the potential role" instead of "develop the role"

**Reply:** Done.

l.49: please correct "The Global Climate Observing System (GCOS) is co-sponsored by the World Meteorological Organization (WMO), the Intergovernmental Oceanographic Commission of the United Nations Educational, Scientific and Cultural Organization (IOC-UNESCO), the United Nations Environment Programme (UN Environment), and the International Science Council (ISC)."

**Reply:** Done.

l.75: please use "discuss the potential role" instead of "develop the role"

**Reply:** Done.

l.89: "in the early 2000s", remove "years"

**Reply:** Done.

l.95: please use "that impact" instead of "which are acting"

**Reply:** Done.

l.100: the listed are not essential climate variables, but the main contributors to the observed warming

**Reply:** We have rephrased the sentence and the figure caption to "the ATMO-RIs cover the main contributors to the observed warming (IPCC, 2021)".

l.114: it is better to say "the most recent IPCC report" rather than "current"

**Reply:** Done.

l.135: remove dot after "forcings."

**Reply:** Done.

l.261 and 163: "At the level"

**Reply:** Done.

l.354-356: the sentence that starts from "In the remaining parts…" can be removed from the text

**Reply:** Done.

l.373: GHGs were defined earlier

**Reply:** Since eth acronym GHG was used only two times in the manuscript, we skipped the use of this acronym and used the full term "greenhouse gases".

l.420: in this context you may want to refer to the World Data Center for Greenhouse Gases run by the Japan Meteorological Agency as an extremely important contribution to the global GHG community in the context of data archiving and dissemination

**Reply:** The reference was implemented. The full sentence is now "Data which were collected before ICOS measurement protocols were put in place, have been secured physically and with respect to data quality. They contribute now to the ICOS record of long-term data sets and to the World Data Center for Greenhouse Gases run by the Japan Meteorological Agency as an extremely important

contribution to the global greenhouse gas community in the context of data archiving and dissemination."

l.435: "covering observations"

**Reply:** Done.

l.475: WMO GHG Infrastructure is now called Global Greenhouse Gas Watch

**Reply:** Corrected.

l.486: remove extra dot

**Reply:** Done.

l.493: ECVs do not vary depending on application because the ONLY application of ECVs is global climate monitoring (by design)

**Reply:** We have removed all references to application areas and focus on climate monitoring.

l.569: either "is illustrated" or "as illustrated", not both

**Reply:** We use "as illustrated".

l.571: "by other scientists"

**Reply:** Corrected.

l.616: "emissions from"

**Reply:** Corrected.

**Referee #2**

The manuscript reads much better with few words used to describe and promote the main points. The manuscript is acceptable for publication after the authors consider the following comments.

- The appendix could be eliminated without significant loss. Afterall this is an Opinion piece which alerts the reader to expect a focused to-the-point narrative. An appendix is antithetical to that expectation. More references/links can mitigate its removal.

- It still feels somewhat repetitive.

**Reply:** As stated in the General Remarks, we prefer to keep the appendix but in an even shorter version. No reference document exists which summarises the structure and benefits of atmospheric research infrastructures. From our understanding, the appendix is needed for readers not that familiar with research infrastructures.

- ln 27 The very long sentence 'In particular…' is particularly hard to read/understand. Suggest dividing in two.

**Reply:** The sentence is divided into two shorter sentences: "In particular, we focus on the role of the atmosphere-centred research infrastructures ACTRIS, IAGOS and ICOS, also referred to as ATMO-RIs, with their capabilities for standardised collection and provision of long-term and high-quality observational data, complemented by rich metadata. The ATMO-RIs provide data through open access and offer data interoperability across different research fields including all fields of environmental sciences and beyond ".

- ln 35 The use of data, environment, observation, and combinations thereof are confusing. For example, the distinction between observations, observational data, and environmental data. Note on ln 44, 'data' has no adjective, but 'observations' is used in the prior sentence. So 'data' should perhaps be 'observational data'. And then in ln46, 'environmental data' is used without obvious distinction. Suggest making it more clear throughout what is being talked about when these words are used.

**Reply:** We clarified the term 'data' as 'observational data'.

- ln 55 'satellite' which is used 8 times is a poor substitute for 'remote sensing' platforms, instruments, observations, data, etc. or 'space-borne', because satellites provide no data, only their instruments do.

**Reply:** Agreed, the inaccurate term 'satellite' was replaced by satellite-borne instrument or space-borne observations, depending on the context.

- ln 74 Another long complex hard-to-understand sentence that would benefit from better syntax or separation into 2 sentences.

**Reply:** The sentence has been split into two shorter sentence and repetitive statements have been removed. The revised section reads now: "In our opinion article, we discuss the potential role that atmosphere-centred ENVRI RIs in Europe can play with their capabilities for standardised collection of observational data, complemented by rich metadata, and their provision through open access. As a result of these capacities for data collection and provision of data and services, we elaborate on the novel research opportunities and methods that arise from the combination of open access and interoperable environmental data with the tools and technologies offered by data-intensive science."

- ln 88 repeat acronym definition.

**Reply:** The sentence has been rephrased and reads now: "In the early 2000s, ESFRI started a roadmap process to establish a comprehensive landscape of pan-European environmental research infrastructures (ESFRI, 2021b)."

- ln 93 acronyms are best defined at first use

**Reply:** The term ATMO-RIs is now introduced in the abstract.

- ln 135. 'forcings,..'

**Reply:** Corrected.

- ln 141 'post-COVID-19'

**Reply:** Corrected.

- ln 142. 'need' for what? 'but' is negating so suggest replacing with 'and'

**Reply:** Replaced as suggested. The sentence reads now: "A number of recent post-COVID-19 studies on the effects of reduced anthropogenic emissions on atmospheric chemical composition (Gkatzelis et al., 2021; Sokhi et al., 2021) impressively illustrate the growing need for, and great potential of, scientific approaches that cross the traditional boundaries of scientific disciplines and methods."

- ln 150 Suggest fixing this sentence as: The discovery of scientific knowledge has developed from experimental sciences and by conducting field observations and experiments; and through theoretical approaches using models and mathematical generalizations to simulate phenomena which are too complex for analytical solutions (Gray, 2009).

**Reply:** Corrected as suggested.

- ln 155. What 'challenge' is being referred to here?

**Reply:** The term "challenges has been replaced by "the challenges of data-driven science".

- ln 158 'always' is hard to defend and unnecessary here so suggest 'often'

**Reply:** We replaced 'always' by 'in the majority of cases'.

- ln 157 'infrastructure'

**Reply:** Corrected.

- ln 191 Better: 'In today's atmospheric sciences and other research areas, scientists are challenged to…'

**Reply:** Corrected as suggested.

- ln 192 'data that are'

**Reply:** Corrected.

- ln 200 Better: 'intensive workflows are capable of exploiting rich and diverse data sources

**Reply:** Corrected as suggested.

- ln 201 What does 'own' mean here?

Reply: 'own' has been replaced by 'new'.

- ln 221 'developing'

**Reply:** Corrected.

- ln 228 Don't agree with 'for machines, knowledge recorded in scientific articles is not accessible'. AI (large language models) can extract and synthesize significant information from published scientific narratives and ideally should be recognized here.

- ln 235 Another very long and complicated sentence. Suggest: 'knowledge in data science. This initiative can fundamentally transform digital scholarship and the way digital infrastructures support scientific knowledge production.'

**Reply to both issues:** The section has been rephrased as suggested and reads now "Initiatives and services such as the Open Research Knowledge Graph (Stocker et al., 2023), are demonstrating advanced technologies like Knowledge Graphs, Natural Language Processing, Terminology Services and Semantic Resources. They also include approaches that integrate the production of FAIR scientific knowledge into data analysis and thus enable the reuse of FAIR scientific knowledge in data science. Recent developments in artificial intelligence are also enabling large language models to extract and synthesise significant information from published scientific narratives. Such initiatives can fundamentally transform digital scholarship and the way digital infrastructures support scientific knowledge production. "

- ln 256 'scientists that will'

**Reply:** Corrected.

- ln 270 Another very long and complicated sentence

**Reply:** The sentence has been restructured to "On the conceptual side, the database accessible to scientists is significantly broadened by the implementation of the science-based framework of ECVs for the provision of ATMO-RI data, combined with the transparent reporting of provenance information in accordance with the FAIR principles, and the discoverability of data via enhanced search capabilities though standardised keywords and vocabularies."

---

## Author Response (AR3)

Response to the remarks from the preceding review file validation

1. If the punctuation marks ". . ." in the section "References" denote some unlisted co-authors, we kindly ask you to change them to the last names and initials. Please see more: https://www.atmospheric-chemistry-and-physics.net/submission.html#references 2.

Reply: The punctuation marks have been removed from the references.

2. It seems that table is included as figure #4. If it is so, it should be re-labelled as table and the references in the manuscript text should be adjusted accordingly. A table may be inserted as an image, but still be called as a table.

Reply: Figure 4 was renamed as Table 1. The respective image file is added.

3. On request by the copyright permitting institution IPCC, the caption to Figure 1 has been adapted to

"Figure 1. Coverage of the main contributors to the observed warming, only the variable "Land-use reflectance and irrigation" is not covered by the ATMO-RIs; the figure is adapted from Figure SPM.2 in IPCC, 2021: Summary for Policymakers (IPCC, 2021b).